# Presence of low virulence chytrid fungi could protect European amphibians from more deadly strains

Mark S. Greener[1], Elin Verbrugghe [1], Moira Kelly[1], Mark Blooi[1], Wouter Beukema [1], Stefano Canessa [1], Salvador Carranza[2], Siska Croubels [3], Niels De Troyer[4], Daniel Fernandez-Giberteau[5], Peter Goethals[4], Luc Lens[6], Zhimin Li[1], Gwij Stegen[1], Diederik Strubbe[6], Robby van Leeuwenberg[1], Sarah Van Praet[1], Mireia Vila-Escale[7], Muriel Vervaeke[8], Frank Pasmans[1] & An Martel [1✉]

Wildlife diseases are contributing to the current Earth's sixth mass extinction; one disease, chytridiomycosis, has caused mass amphibian die-offs. While global spread of a hypervirulent lineage of the fungus *Batrachochytrium dendrobatidis* (*Bd*GPL) causes unprecedented loss of vertebrate diversity by decimating amphibian populations, its impact on amphibian communities is highly variable across regions. Here, we combine field data with in vitro and in vivo trials that demonstrate the presence of a markedly diverse variety of low virulence isolates of *Bd*GPL in northern European amphibian communities. Pre-exposure to some of these low virulence isolates protects against disease following subsequent exposure to highly virulent *Bd*GPL in midwife toads (*Alytes obstetricans*) and alters infection dynamics of its sister species *B. salamandrivorans* in newts (*Triturus marmoratus*), but not in salamanders (*Salamandra salamandra*). The key role of pathogen virulence in the complex host-pathogen-environment interaction supports efforts to limit pathogen pollution in a globalized world.

[1] Wildlife Health Ghent, Department of Pathology, Bacteriology and Avian Diseases, Faculty of Veterinary Medicine, Ghent University, B-9820 Merelbeke, Belgium. [2] Institute of Evolutionary Biology (CSIC-UPF), 08003 Barcelona, Spain. [3] Department of Pharmacology, Toxicology and Biochemistry, Faculty of Veterinary Medicine, Ghent University, B-9820 Merelbeke, Belgium. [4] Department Animal Science and Aquatic Ecology, Faculty of Bioscience Engineering, Ghent University, Coupure Links 653, B-9000 Gent, Belgium. [5] Centre de Recerca i Educacio Ambiental de Calafell (GRENP - Ajuntament de Calafell), 43882 Segur de Calafell, Spain. [6] Terrestrial Ecology Unit, Department of Biology, Ghent University, K.L. Ledeganckstraat 35, 9000 Ghent, Belgium. [7] Oficina Tècnica de Parcs Naturals, Diputació de Barcelona, 08036 Barcelona, Spain. [8] Agentschap voor Natuur en Bos, Havenlaan 88, 1000 Brussel, Belgium. ✉email: An.Martel@ugent.be

Global invasion by highly virulent chytrid fungi has resulted in the most extensive disease-driven loss of biodiversity ever recorded, with the decline of 500 amphibian species worldwide[1,2]. The skin disease chytridiomycosis is caused by two fungal species, *Batrachochytrium dendrobatidis* (*Bd*)[3] and *B. salamandrivorans* (*Bsal*)[4]. The vast majority of chytridiomycosis-driven declines have been attributed to the hypervirulent and panzootic lineage of *Bd* (*Bd*GPL)[1,5,6]. Despite its presumed high virulence and global distribution, amphibian community declines caused by *Bd*GPL are predominantly limited to disease hotspots in North, Central and South America, Australia, Europe and Africa[1,2,7–9]. Since *Bd*GPL is also widely distributed in regions not affected by declines[1], its mere presence is a poor predictor of disease dynamics. In Europe, *Bd*GPL impacts vary from regional mortality events and declines in a limited number of species[7,10–12] to widespread endemism characterized by host–pathogen co-existence in a broad range of host species[1,2,13,14]. Current knowledge of the host–pathogen–environment interaction[15–22], notably the role of local *Bd*GPL virulence types, offers an incomplete explanation of these different infections and disease dynamics.

Here we predict that *Bd*GPL co-existence with amphibian communities is mediated by low virulence of the present *Bd*GPL isolates and explore the extent to which low-virulence isolates confer protection against hypervirulent *Bd* or *Bsal*. We determined the occurrence of *Bd*GPL in native amphibian populations in Belgium. We then compared pathogen virulence traits of northern European *Bd*GPL isolates to those of a known hypervirulent isolate. Finally, we estimated the impact of pre-existing infections with less virulent *Bd*GPL isolates on virulent *Bd*GPL and *Bsal* infections.

## Results

**Populations persist in the presence of *Bd*GPL.** Opportunistic sampling of 1483 amphibians for *Bd* in 2015–2016 revealed the presence of *Bd*GPL in 17 out of 63 locations and 5 out of 7 species sampled in Flanders, Belgium (Supplementary Fig. 1). Prevalence ranged between 0 and 35% per species (Supplementary Table 1). Besides, 25 cases of amphibian mortality, comprising 47 animals belonging to seven species (1 *Rana temporaria*, 20 *Bufo bufo*, 2 *Triturus cristatus*, 11 *Ichthyosaura alpestris*, 7 *Lissotriton vulgaris*, 3 *Salamandra salamandra*, 3 *Lissotriton helveticus*) were reported to the regional hotline (passive surveillance system) and submitted for postmortem analysis during a 4-year period (2015–2018). Chytridiomycosis was not diagnosed in any of these reported cases.

The impact of *Bd* on native amphibians was estimated both in remnant populations of a threatened chytridiomycosis susceptible species (midwife toads, *Alytes obstetricans*) and in populations of a common newt considered to be a *Bd* supershedder (alpine newt, *Ichthyosaura alpestris*). During the 4-year study, the midwife toad populations persisted in the presence of *Bd* infections (Supplementary Table 2). *Bd* occurrence was observed in 10 of 26 study populations of alpine newts and did neither result in reduction of newt abundance ($F_{(2,76.2)} = 0.04$, $p = 0.96$) nor in reduction of body condition (i.e. scaled mass index, $F_{(1,1728.13)} = 1.66$, $p = 0.20$) (Supplementary Table 3, Supplementary Fig. 2).

**Higher fecundity in hypervirulent *Bd*GPL compared to local *Bd*GPL isolates.** Ten *Bd*GPL isolates were cultured from midwife toads (*A. obstetricans*), alpine newts (*I. alpestris*) or invasive bullfrogs (*Lithobates catesbeianus*) across Flanders, Belgium (Supplementary Table 4). These isolates were confirmed to belong to the GPL clade using a lineage-specific PCR[23] and this was corroborated by whole-genome sequencing of 6 isolates[1]. Their culture phenotype (growth characteristics and morphological

traits) was compared to that of the hypervirulent *Bd*JEL423, isolated from an episode of amphibian mortality in the neotropics[24]. After 7 days of culturing, the hypervirulent *Bd*JEL423 had significantly larger sporangia (linear model $\beta = 0.28 \pm 0.02$, $t(10,416.8) = 14.9$, $P < 0.001$; pairwise Tukey's HSD contrasts JEL423-other isolates $P < 0.01$; Fig. 1) and produced significantly more zoospores (linear model $\beta = 335.4 \pm 26.4$, $t(10,32) = 12.72$, $P < 0.001$; pairwise Tukey's HSD contrasts JEL423-other isolates $P < 0.01$; Fig. 1) than any of the local *Bd* isolates (Fig. 1), corroborating previous reports that zoospore production and sporangium size are linked to pathogen virulence[25,26]. Although significant variation in phenotypic traits was also noticed among the local isolates, multivariate ordination of phenotypic traits grouped *Bd*JEL423 apart from all other isolates (Fig. 1).

**Midwife toads are a European sentinel species for *Bd*-induced chytridiomycosis.** To assess the potential impact of invasion of hypervirulent *Bd*GPL in the broader amphibian community, we compared susceptibility of seven indigenous amphibian species to exposure to the hypervirulent *Bd*GPL isolate *Bd*JEL423 and the local isolate *Bd*BE1. In contrast to the midwife toads that died after exposure to *Bd*JEL423, European tree frogs (*Hyla arborea*), fire salamanders (*Salamandra salamandra*), spadefoot toads (*Pelobates fuscus*), alpine newts (*I. alpestris*) and natterjack toads (*Epidalea calamita*) became infected with both isolates (Supplementary Fig. 3) but did not develop lethal chytridiomycosis within the 13 weeks of the experiment. Throughout the trial, great crested newts (*Triturus cristatus*) and common toads (*Bufo bufo*) never tested positive for *Bd* (Supplementary Fig. 3). The marked susceptibility of midwife toads and the low susceptibility of fire salamanders and common toads correlate with marked fungal growth in mucosomes collected from midwife toads and with fungal growth inhibition by mucosomes collected from fire salamanders (Fig. 2a, Supplementary Fig. 4) and reflects the European field situation (endemic and widespread presence of *Bd*, with focal outbreaks)[1,7,15,16,27–29]. Based on these results we classify the midwife toad as a *Bd* sentinel (canary in the coal mine for virulent *Bd* infections in Europe) species in Europe and decided to use this species in further experiments to test the pathogenicity of local isolates.

**Midwife toads tolerate infection with local *Bd*GPL isolates but are highly susceptible to the hypervirulent *Bd*GPL isolate JEL423.** Compared to the hypervirulent *Bd*GPL isolate *Bd*JEL423, all four local *Bd*GPL isolates included were low pathogenic in an infection trial using recently metamorphosed midwife toads (*Alytes obstetricans*) (Fig. 3). *Bd*JEL423 caused 100% mortality (9 out of 9 animals, housed individually) with a mean time to death after exposure of 27 days (range 16–64). Of the 29 toads infected with the local *Bd*GPL isolates, only one died due to infection. The probability of death was not significantly different from the uninfected control group (for penalized GLM coefficients, all $z$ (5,43) < 0.7, $p > 0.5$; Fig. 3, Supplementary Table 5).

**BdGPL pathogenicity correlates with Crinkler and Necrosis (CRN) gene expression pattern and host cell invasion.** Searching for pathogenicity determinants, the isolates were further compared by analysing the expression of a selection of *Bd* virulence genes in spores and spores that were first exposed to midwife toad skin. Regarding virulence gene expression, marked inter-isolate variation was noticed. Distinctive features mainly pertained to the expression of CRN and metalloprotease genes (Supplementary Tables 6–7). Expression of CRN genes has been linked to the early infection stage of *Bd*JEL423[5], and was low in spores of local *Bd*GPL isolates (*Bd*BE1, *Bd*BE3, *Bd*BE4 and

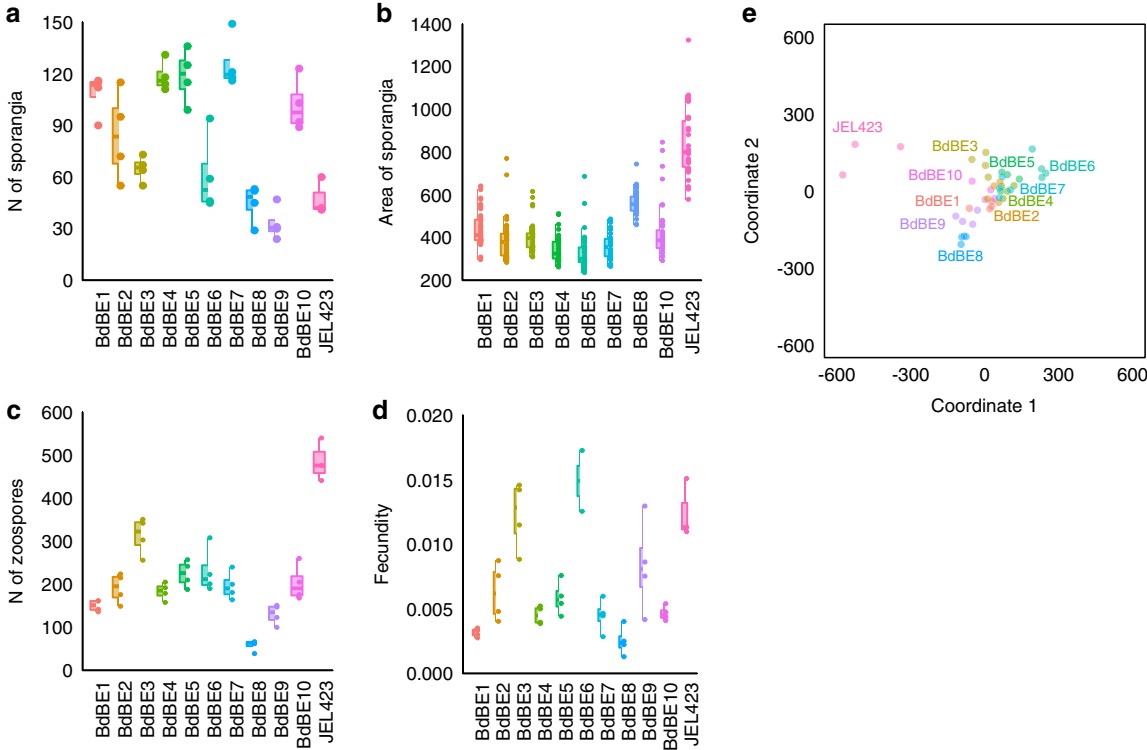

**Fig. 1 Experimental results for phenotypic traits of *B. dendrobatidis* isolates. a–d** Boxes indicate 25th and 75th percentiles, central lines the mean, bars the 95th percentiles, and points indicate individual samples. N of sporangia = number of sporangia in the central 1000 × 1000 pixels, $n = 4$ independent experiments; area of sporangia = area of the largest 10 sporangia, $n = 10$ technical replicates per independent experiment; N of zoospores = number of spores in the central 1000 × 1000 pixels of the well/image, $n = 4$ independent experiments; fecundity was calculated following the formula (average $N_{ZOOSPORE}$/average $N_{SPOR}$)/average $A_{SPOR}$, $n = 4$ independent experiments. **e** Non-metric multidimensional scaling across the four variables in (**a–d**); labels indicate one point per group as a colour legend. *Bd*BE1-10 are local isolates isolated in this study. JEL423 is a *Bd*GPL isolate isolated from an episode of amphibian mortality in the neotropics[16]. Source data are provided as a Source data file.

*Bd*BE5) compared to the hypervirulent *Bd*JEL423. Incubation with midwife toad tissue resulted in increased CRN gene expression for isolates *Bd*JEL423, *Bd*BE1 and *Bd*BE3 and decreased expression for *Bd*BE4 and *Bd*BE5 (Fig. 2b, Supplementary Fig. 5, Supplementary Tables 6–7). In vitro, this pattern of CRN gene expression was correlated with invasiveness and intracellular growth in amphibian A6 cells. Intracellular growth, characterized by the formation of intracellular daughter thalli, was only observed for *Bd*JEL423, *Bd*BE1 and *Bd*BE3. Growth of *Bd*BE4 and *Bd*BE5 was dominated by an epibiotic growth, limited to *Bd* development outside the host cells (Fig. 2d, e, Supplementary Figs. 6–8). Relative CRN gene expression followed the same pattern as colonisation capacity in the above mentioned in vivo infection experiment, with significant correlation between CRN gene expression of fresh spores of local *Bd*GPL isolates relative to *Bd*JEL423 and skin colonisation capacity (CRN_22492 $r_s = 0.88$, $p \le 0.05$, $n = 38$) (CRN_25085 $r_s = 0.73$, $p \le 0.05$, $n = 38$) (CRN_23176 $r_s = 0.88$, $p \le 0.05$, $n = 38$)[5].

**Genome-wide variance associated with *Bd*GPL virulence and host cell invasion.** Whole-genome sequence data were mapped against the reference genome for *Bd*JEL423 to identify genome-wide variation associated with the low virulence (*Bd*BE1, *Bd*BE3, *Bd*BE4 and *Bd*BE5) and cell invasion (*Bd*BE1, *Bd*BE3 and *Bd*JEL423) phenotypes. Annotation of variants predicted to be highly deleterious or disruptive to gene function predicted 101 genes presenting differentiating variants between local isolates and the hypervirulent *Bd*JEL423 isolate and 64 genes differentially affected between isolates displaying the invasive versus the epibiotic phenotype. Many of these genes were annotated as

candidate Crinkler genes (12 genes), or contained common effector protein domains such as SigP4 secretory signals (28 genes), transmembrane domains (10 genes), or metalloprotease (4 genes), aspartate proteases (12 genes) and CBM18 chitin-binding PFAM domains (4 genes, Supplementary Datasheet 1). Fisher's exact test indicated gene enrichment of five PFAM domains in both sets of affected genes, including Aspartate proteases, the chitin-binding CBM18 domain and the M36 metalloprotease-associated FTP domain (Supplementary Datasheet 1). Three of the genes affected by deleterious mutations in all local isolates show upregulated expression in *Bd*JEL423 in vivo[5], including BDEG_22285, a secreted M36 metallopeptidase. The established role of Crinkler proteins[30], M36 metalloproteases[31,32] and aspartate proteases[33–35] in fungal pathogenesis suggest these may be *Bd* effector proteins that are differentially associated with function-altering mutations in virulent and invasive phenotypes, and highlights targets for future investigation. As previous studies have found pathogen virulence attenuation in *Bd*GPL to be associated with an overall decrease in copy number[24], this was compared in the local and *Bd*JEL423 isolates. While the local isolates showed extensive copy number variation of large sequence regions up to entire supercontigs, no such overall decrease in copy number was observed in the local isolates compared with *Bd*JEL423 (Supplementary Fig. 9).

**Local *Bd*GPL protects midwife toads against hypervirulent *Bd*GPL.** To assess the level of protection conferred by the local *Bd*GPL isolates against invasion by hypervirulent *Bd*GPL, we exposed the animals that were previously infected with one of four local *Bd*GPL isolates to the hypervirulent *Bd*JEL423 and

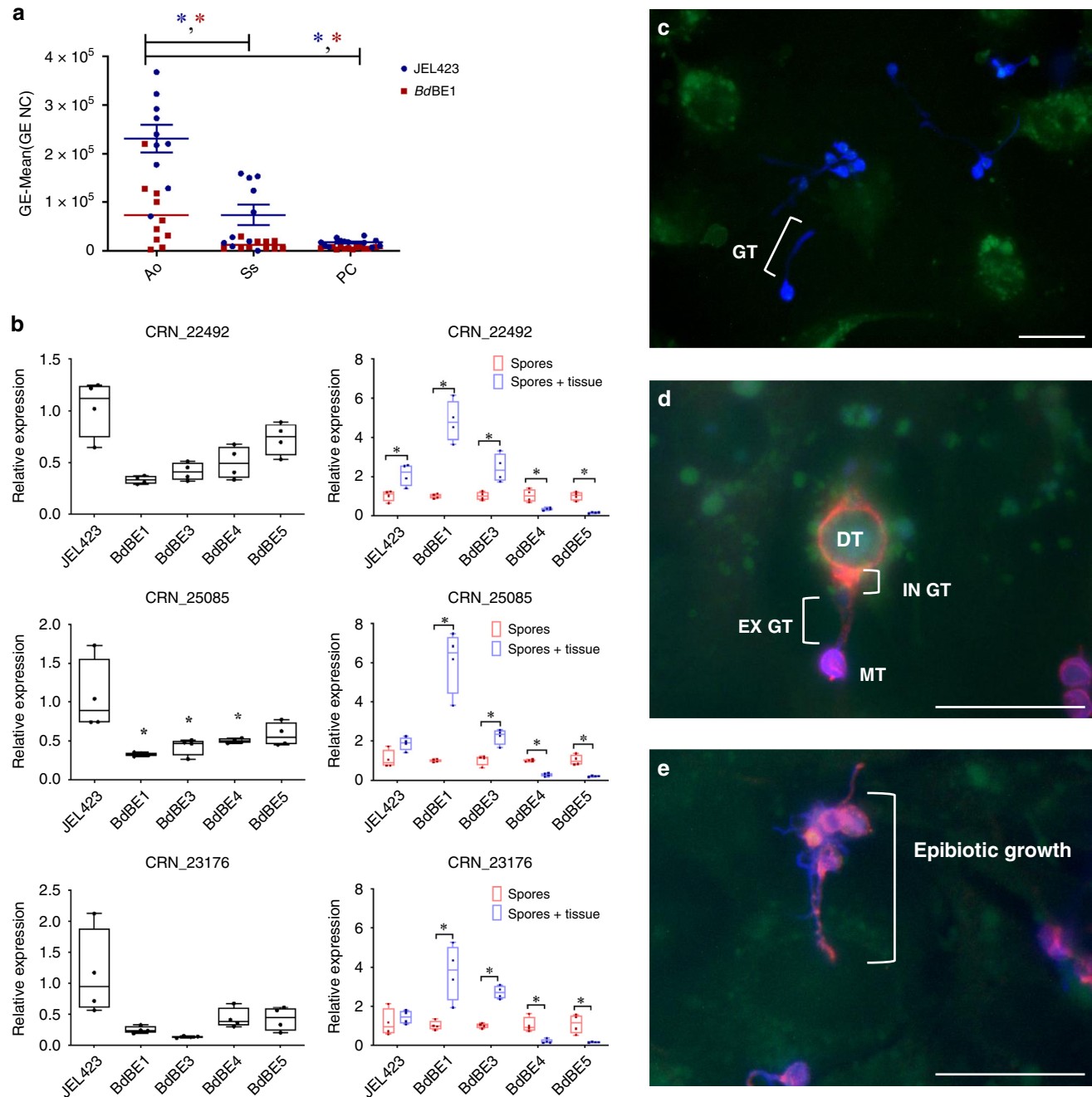

compared infection dynamics and disease with that of age-matched, naive toads exposed to *Bd*JEL423 only. Animals that had been previously infected with the *Bd*BE1 and *Bd*BE3 isolate showed significantly higher survival than those infected directly with *Bd*JEL423 (GLM $\beta = -2.59 \pm 1.29$, $z(2,32) = -2.01$, $p = 0.044$; Fig. 3, Supplementary Table 8).

In a final proof of concept study, infection dynamics of the hypervirulent *Bd*JEL423 were compared between midwife toadlets that were either pre-exposed or not pre-exposed (naive) to low-virulence *Bd*BE3 ($n = 20$ per treatment, all animals kept individually) 4 weeks before *Bd*JEL423 challenge. Exposure to *Bd*BE3 resulted in infection in all animals in the absence of mortality and a median peak load of 140 GE (range 14–1164 GE). Subsequent challenge of these pre-exposed toads with *Bd*JEL423 resulted in detection of low *Bd* levels in only two toadlets (3 and 23 GE) and only on the first sampling occasion. In contrast, all naive animals developed significant infection loads during the 4-

week follow-up period after challenge with *Bd*JEL423 (median peak load 1802 GE, range 58–19,100 GE) (Fig. 3). Prior infection with the low-virulence *Bd*BE3 thus resulted in less infections upon challenge with the highly virulent *Bd*JEL423 (Fisher's exact test, two-tailed *P* value < 0.0001).

**BdGPL alters virulent Bsal infection in marbled newts, but not fire salamanders.** Local *Bd*GPL temporarily colonized fire salamanders (*S. salamandra*), marbled newts (*Triturus marmoratus*) and ribbed newts (*Pleurodeles waltl*) without causing clinical signs or mortality (Fig. 4). The effect of pre-exposure to local *Bd*GPL on the course of a subsequent *Bsal* infection depended on the infected species (Fig. 4, Supplementary Table 9). In fire salamanders, exposure to local *Bd*GPL had no effect on *Bsal*-induced mortality (penalized GLM $\beta = 0.00 \pm 2.17$, $z(2,26) = 0.00$, $p = 1$) or infection course (parametric survival regression $\beta = -0.07 \pm$

**Fig. 2 BdGPL virulence coincides with growth ability in mucosomes, Crinkler (CRN) gene expression and adhesion and invasion capacities. a** Growth (expressed in corrected GE values) of BdBE1 and BdJEL423 in mucosomes collected from different amphibian species. PC: positive control; Ao: *Alytes obstetricans*; Ss: *Salamandra salamandra*; NC: negative control. Significant differences are shown with an asterisk and were assessed by a Kruskal–Wallis analysis, followed by pairwise Mann–Whitney U-test (two-tailed) with a Bonferroni-corrected P value of 0.017 (Ao vs Ss: P value = 0.001 (BdJEL423) and 0.009 (BdBE1); Ao vs PC: P value < 0.001 (BdJEL423) and 0.003 (BdBE1)). The experiment was carried out in tenfold and individual data points are shown (blue = BdJEL423; red = BdBE1), with the mean ± s.e.m. depicted by the horizontal bars. **b** Box plots of mean fold changes in mRNA expression of CRN genes (CRN_22492, CRN_25085 and CRN_23176). The data in the left panel show the normalized target gene amount in spores of each isolate ($n = 4$) relative to spores of BdJEL423 ($n = 4$) which is considered 1. The data in the right panel show the normalized target gene amount in spores that were incubated with skin tissue of *A. obstetricans* for 2 h ($n = 4$) relative to spores of the respective isolate ($n = 4$) which is considered 1. Boxes indicate 25th and 75th percentiles, central lines the median, bars the minima and maxima, and points indicate individual samples. Target genes were based on Farrer et al.[5]. Left panel: an asterisk indicates a significant difference compared to BdJEL423 spores (Kruskal–Wallis analysis, followed by pairwise Mann–Whitney U-tests (two-tailed) with a Benjamini–Hochberg-adjusted P value < 0.05). Right panel: an asterisk indicates a significant difference compared to spores of the respective isolate (Kruskal–Wallis analysis, followed by pairwise Mann–Whitney U-tests (two-tailed). Individual P values are shown in Supplementary Tables 6–7. **c** Representative image showing germ tube formation of BdJEL423. Fluorescent signals of Bd (Calcofluor White (blue)) and A6 cells (green cell tracker) were merged to assess the ability of germ tube formation, 4 h after contact with the cells. **d, e** Representative images demonstrating endobiotic growth (**d**: BdJEL423) and epibiotic growth (**e**: BdBE4) of Bd. The cell wall of extracellular Bd was coloured using Calcofluor White (blue). Bd was visualized using Alexa Fluor 568 targeting a polyclonal antibody against Bd (Thomas et al.[62]), resulting in red fluorescence of both intracellular and extracellular Bd. The Bd-exposed A6 cells were stained with a cell tracker (green). **d** New intracellular chytrid thalli are formed and the cell content of the mother thallus is transferred into the new daughter thallus (DT). **e** Epibiotic growth is limited to Bd development outside the host cells. GT germ tube, EX GT extracellular germ tube, IN GT intracellular germ tube, MT mother thallus, DT daughter thallus. Scale bar = 20 μm. Three independent in vitro experiments were conducted with every condition being tested in triplicate, with similar results. Source data are provided as a Source data file.

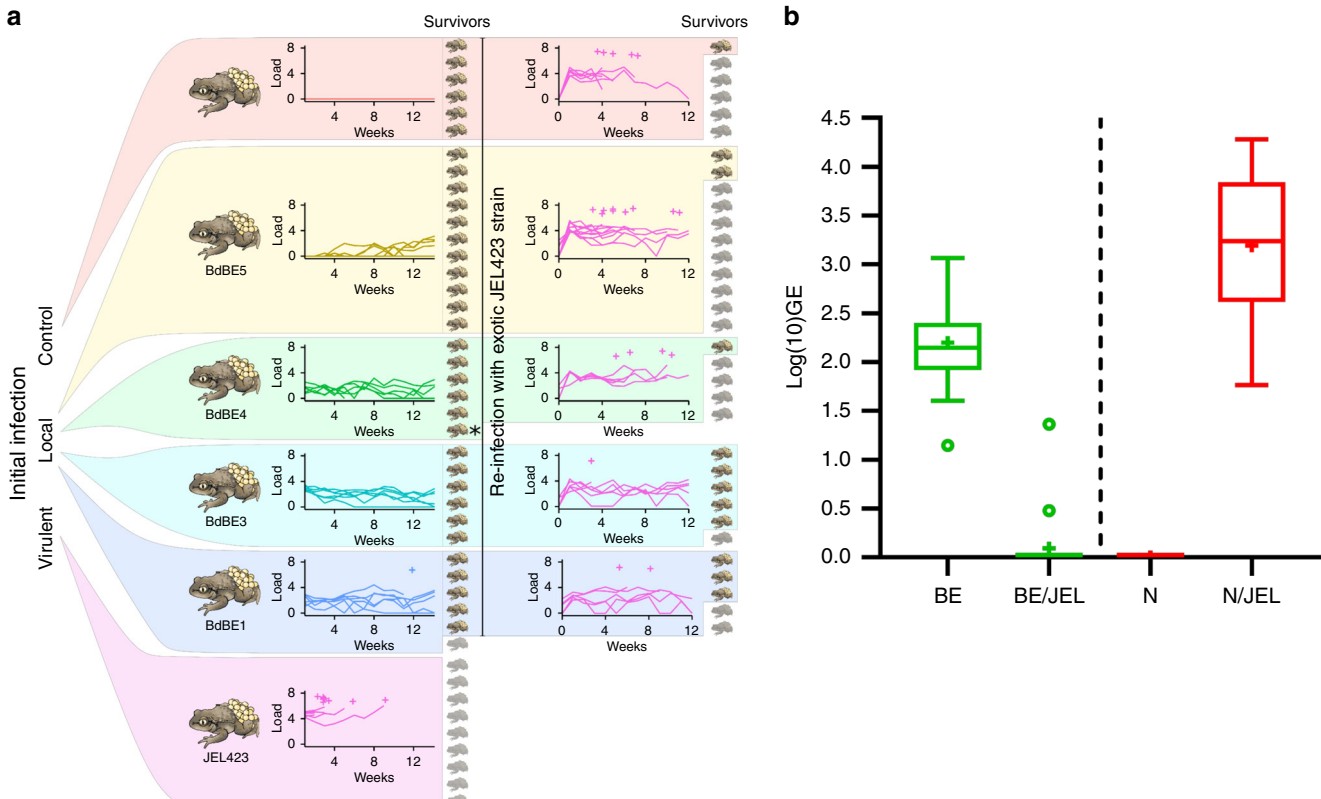

**Fig. 3 Local BdGPL isolates protect midwife toads against infection with virulent BdGPL. a** (Left) Toads exposed to local (BdBE) isolates had survival comparable to the control (non-exposed) group; toads exposed to hypervirulent BdJEL423 had 100% mortality. (Right) When all survivors were re-infected with BdJEL423, toads previously exposed to BdBE1 and BdBE3 had higher survival than naive toads (control) or toads previously exposed to BdBE4-5. For each group, curves indicate infection load ($\log_{10}[\text{GE}+1]$), crosses indicate times of individual deaths (crosses offset vertically for readability, dead individuals shaded). The asterisk * indicates one animal that died of non-treatment-related causes between the two experiments. **b** Tukey-style box plots of Bd peak loads in midwife toads either pre-exposed (BE: green) or not pre-exposed (N: red) to the low-virulence BdBE3 isolate and 4 weeks later challenged with the highly virulent BdJEL423 (BE/JEL and N/JEL). $n = 20$ biologically independent animals per condition. Source data are provided as a Source data file.

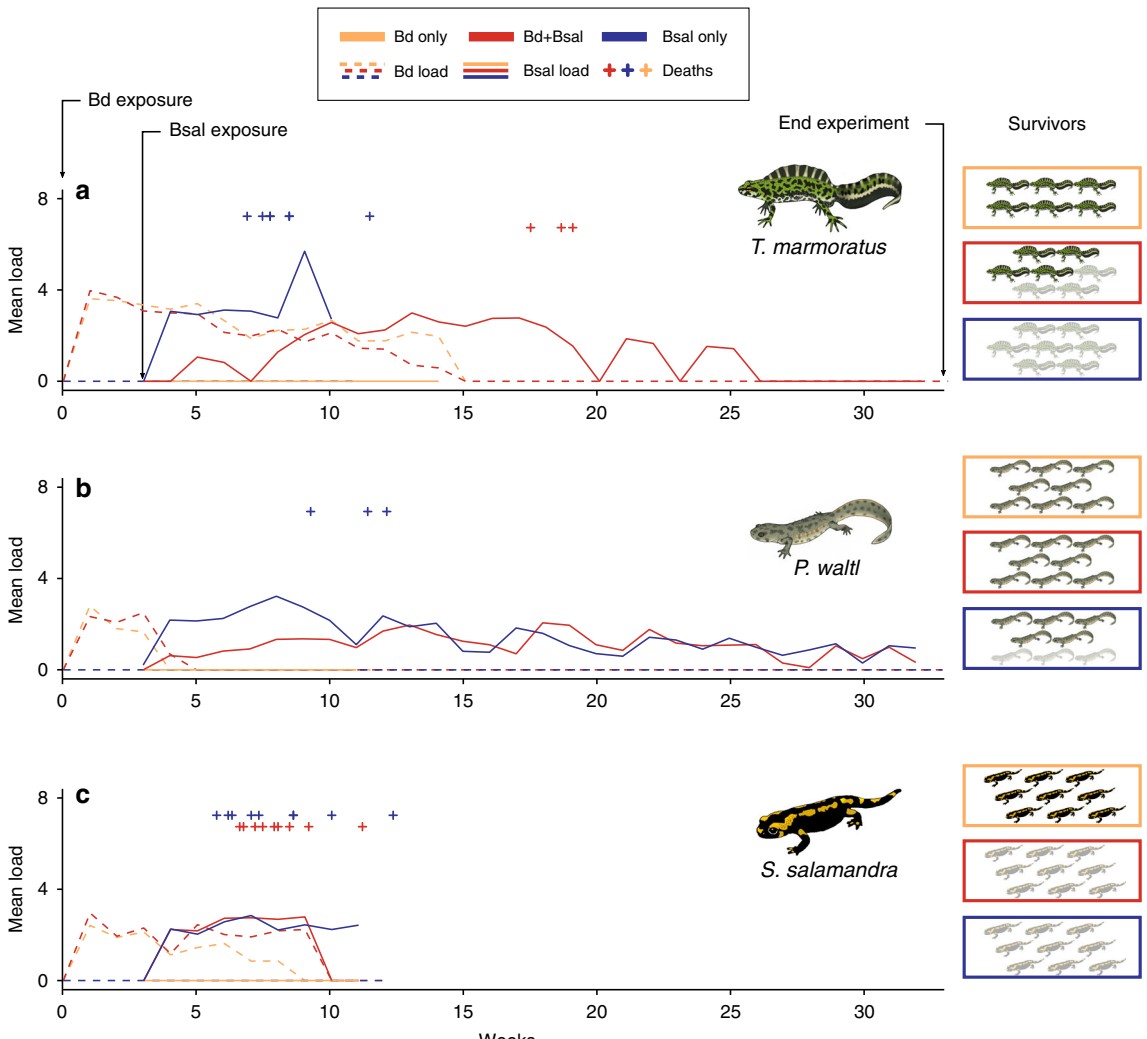

**Fig. 4 Low-virulence *Bd*GPL isolates protect some urodelan species, but not others, against infection with *B. salamandrivorans*. a** For *Triturus marmoratus*, low virulent BdGPL exposure reduced *Bsal*-induced mortality but increased the length of the infectious period. **b** For *Pleurodeles waltl*, low-virulence BdGPL exposure reduced *Bsal*-induced mortality but had no effect on the length of infectious period or on infection loads. **c** For *Salamandra salamandra*, low virulent BdGPL exposure had no effect on *Bsal*-induced mortality or infection course. For each species, *Bsal* exposure occurred 21 days after *Bd* exposure. For all groups, curves indicate infection load ($\log_{10}$[GE+1]), crosses the time of individual deaths (offset vertically for readability), boxes the number of survivors in each treatment group by species (dead individuals shaded). Source data are provided as a Source data file.

0.15, $z(1,18) = -0.43$, $p = 0.67$); *Bsal* exposure always resulted in lethal infection at an average of 35 days for both treatments. In ribbed newts, *Bd*GPL exposure did not significantly change mortality (penalized GLM $\beta = 0.00 \pm 2.18$, $z(2,23) = 0.00$, $p = 1$) or the duration of the infection period (model could not be fitted; Fig. 4b), yet, mortality was observed in the non-*Bd* pre-exposed newts only. All marbled newts that were inoculated with *Bsal* only died within an average of 37 days after exposure, whereas four out of seven newts survived when pre-exposed to *Bd*GPL (penalized GLM $\beta = 2.95 \pm 1.74$, $z(2,19) = 1.71$, $p = 0.088$, one-sided *t*-test, $p = 0.037$). However, *Bd*GPL pre-exposure increased the duration of the infectious period (parametric survival regression $\beta = 1.49 \pm 0.21$, $z(1,14) = 7.14$, $P < 0.001$) and it took on average 107 days before three of the newts died.

## Discussion

Although *Bd*GPL has caused death and species extinction on a global scale, its impact in Europe is currently limited to specific regions. Our results suggest that low-virulence *Bd*GPL isolates co-

occur with amphibian communities in the absence of population declines and that only a limited number of European amphibian species may be susceptible to the highly virulent isolates associated with massive declines elsewhere. Results of our laboratory trials with seven indigenous amphibian species largely match the in situ condition in northern Europe[1,13,15,27–29]. None of the tested species was susceptible to *Bd*-induced chytridiomycosis after exposure to the local *Bd*GPL isolates but several showed infection patterns that corroborate their potential as a chytrid reservoir. Midwife toads persisted in small and isolated populations during at least 4 years in the presence of *Bd* and *Bd* infection could not be linked to decreased body condition or host abundance in alpine newts. This finding corroborates the current co-existence of low-virulence *Bd*GPL with northern European amphibian assemblages, despite the occurrence of erratic mortality[13,36].

In contrast, the hypervirulent *Bd*GPL isolate did cause lethal infections in a single amphibian species, the midwife toad, while several other species showed relatively high-level long-term infections. These experimental results agree with *Bd* epizootics

and population declines observed in midwife toads in Southern Europe[7,10–12,37].

Our results suggest that if a hypervirulent *Bd*GPL similar to the one used in this study invaded our study area, two scenarios might occur. If the invaded host population is entirely *Bd*-naive, susceptible hosts (e.g. midwife toads) are likely to be lethally affected. Co-occurring, tolerant hosts (e.g. alpine newts) may sustain high-level infections and act as pathogen reservoirs. If spatio-temporal factors shape a conducive environment, these disease dynamics might result in mass die-offs, population declines and even extirpation of the susceptible species[38]. However, pre-existing low-virulence *Bd* isolates may at least initially affect disease dynamics of an invading hypervirulent *Bd*. Their protective effect against disease is associated with several traits that correlate with their colonisation ability and invasiveness: expression and function of *Bd* crinkler genes, metalloproteases and other effector proteins. However, the hypervirulent isolate's high fecundity, marked invasiveness in amphibian cells and more pronounced colonization capacity in vivo suggest a distinct competitive advantage over the local isolates[39]. Moreover, co-infections may result in super-infection and/or hybridisation[3,40,41], with highly unpredictable outcomes.

Reduced mortality and delayed time to death demonstrate that prior exposure to *Bd* reduces pathogenicity of *Bsal* in marbled newts. This may equally pertain to the ribbed newts, in which *Bsal* mortality occurred only in newts that were not previously exposed to *Bd*. However, low mortality rate and relatively small sample size hamper meaningful statistics for this species. In contrast, all fire salamanders succumbed to *Bsal* infection within a similar timeframe, regardless of prior *Bd* infection. The host species-dependent protective effect of a pre-existing *Bd* infection against *Bsal* may present as a double-edged sword for the amphibian community. The reduced pathogenicity in the marbled newts coincides with a longer infectious period and high *Bsal* infection loads, which is likely to facilitate *Bsal* transmission to susceptible animals. Increased transmission opportunities may offer an additional explanation for the local extirpations observed in *Bsal*-infected fire salamander populations in parts of Europe where *Bd* is widespread[4,42]. In amphibian communities, simultaneous introduction of both pathogens may be subject to different disease dynamics. Simultaneous co-infections with *Bd* and *Bsal* in American newts (*Notophthalmus viridescens*) resulted in disease exacerbation[43].

In conclusion, our results show a complex disease landscape with multiple implications for biodiversity conservation. First, the mere demonstration of *Bd*GPL presence in amphibian communities is not sufficient to predict disease impacts on susceptible species. Second, co-existence of amphibian communities with low-virulence *Bd*GPL may alter the outcome of an invasion by a hypervirulent isolate[44]. Protection may increase the probability of host populations surviving a hypervirulent chytrid incursion, but also amplify reservoir dynamics and increase the risk to highly susceptible species. European mitigation programmes could be fine-tuned to focus on the highly virulent isolates, but this would require the availability of a rapid diagnostic pathogen virulence assay.

## Methods

**Impact of *Bd* endemism in amphibian communities in Flanders**. In a first study, *Bd* prevalence was determined across our study area (Flanders, Belgium). We sampled 1483 amphibians belonging to 62 populations in 2015–2016 (Supplementary Fig. 1). To detect the presence of *Bd*, we collected swabs from the superficial skin surface of metamorphosed animals or the mouthparts of larval anurans. To study potential co-existence of *Bd* in the study region with small populations of a susceptible species (where negative effects are expected to be most obvious), in a second study, we sampled five breeding sites of midwife toads in Flanders for 4 consecutive years (Supplementary Table 2). In these breeding sites, larvae were

counted once a year and their mouthparts were sampled for the presence of *Bd*. In a third field study, we selected 26 ponds across our study area containing at least a population of alpine newt (*Ichthyosaura alpestris*), being the European urodele most likely infected by *Bd*. Ponds were sampled with funnel traps three or four times (depending on the presence of water) with a 1-month interval (March–June 2019). An envisaged 30 newts per sampling per pond were swabbed for the presence of *Bd* (Supplementary Table 3), weighed to the nearest 0.1 g and the snout-vent length measured to the nearest mm. As an estimate of body condition, we used the scaled mass index (SMI)[45], which adjusts the mass of all individuals to that which they would have obtained if they had the same body size. SMI was calculated using the equation of the linear regression of log-body mass on log-snout-vent length estimated by type 2 (standardized major axis; SMA) regression[45]. Eleven outliers were present (i.e. |standardized residual| >3). These observations were not used for deriving SMI relationships (as per Peig and Green[45]). The regression slope was 2.87, and average snout-vent length was 42.7 mm. We thus calculated the SMI as (body mass × (42.7/snout-vent length)$^{2.87}$). The average number of individuals caught per fyke per pond was used as proxy for newt density. To test whether trends in newt density (i.e. average number of newts per fyke) differed between *Bd* positive and *Bd* negative ponds, a generalized linear mixed model (GLMM) was used specifying newt density as the dependent variable and the interaction between time (month) and *Bd* status (positive versus negative) as independent variables. Trends in newt density were better approximated by a quadratic relationship compared to a linear trend (delta AIC = 5.73). To test whether trends in SMI differed between *Bd* positive versus *Bd* negative newts, a GLMM was implemented using SMI as the dependent variable and time (month), *Bd* status (positive versus negative), newt sex (male versus female) and newt density (see above) as independent variables. The initial model contained all two-way interactions between the independent variables. For both GLMMs, pond was implemented as a random factor, and a Gaussian error structure was specified (model residuals were normally distributed, Shapiro–Wilk $W > 0.90$). A frequentist approach was adopted whereby initial models were reduced in a stepwise manner, by excluding the variable with the highest $P$ value until only $P < 0.05$ predictors remained.

This field study was performed with approval of the Flemish government (derogation number ANB/BL/FF-V15-00015).

Animals were handled while wearing a fresh pair of non-powdered, disposable gloves. Equipment and field clothing were cleaned and disinfected between visits to sampling locations.

Detection of chytrid associated disease in the study region was done by postmortem examination of field cases of amphibian disease or mortality over a period of 4 years. Wildlife Health Ghent hosts an amphibian disease hotline where suspect cases of infectious disease are submitted (cases with obvious traumatic causes such as predation or traffic are not withheld). The dead amphibians are routinely examined for the presence of *Bd*, *Bsal* and Ranavirus using (q)PCR[46–48].

**Isolate collection**. Isolates were obtained from wild amphibians collected July 2015–September 2016 in the Flanders region of Belgium. Isolation was carried out using the protocol described in Fisher et al.[49]. Using a sterile needle, small skin sections 1–2 mm were cleaned of surface-contaminating bacteria and fungi by dragging it through agar-medium. Subsequently, the tissue sample was transferred into a sterile TGhL medium with antibiotics (200 mg/L penicillin-G and 400 mg/L streptomycin sulphate) and incubated at 20 °C. Six isolates were genotyped and their topology depicted in the global phylogeny in O'Hanlon et al.[1]. These isolates belong to three clades. Full information on each isolate can be found in (Supplementary Table 4). All isolates were preserved in liquid nitrogen at passage 10. Before use in an experiment, isolates were passaged one time in TGhL broth at 20 °C.

**Hygiene and biosafety protocols**. The animal experiments were performed under strict BSL2 conditions. During the fieldwork, each individual was handled with a new pair of nitrile gloves. At the end of each field visit, boots, dipnets, funnel traps and other equipments were disinfected with a 1% Virkon® solution for at least 5 min[50].

**Phenotypic characterisation of local *Bd* isolates**. All cultures (*Bd*BE1, 2, 3, 4, 5, 6, 7, 8, 9, 10 and *Bd*JEL423) were grown on TGhL agar at 20 °C, for 5 days. Plates were washed with sterile distilled water, and the water containing the zoospores was passed over a sterile mesh filter with pore size 10 mm (Pluristrainer, PluriSelect). Using a haemocytometer and inverted microscope (Nikon Eclipse TS100, Nikon) the spores were counted. *Bd* spores ($1 \times 10^6$ spores per well) were seeded in TGhL medium (1.6% tryptone, 0.4% gelatin hydrolysate and 0.2% lactose in $H_2O$) supplemented with 50% $H_2O$ into 24-well tissue culture plates. Wells were photographed using an Olympus CKX41 with an attached camera (Olympus SC50, Olympus) with a (2560 × 1920) pixel field, ~72, 96, 120, 144 and 168 h after inoculation. The experiment was performed in quadruplicate. Images were analysed using ImageJ 1.52d software, with the following measurements taken: (1) $N_{ZOOSPORE}$ = number of spores in the central 1000 × 1000 pixels of the well per image, (2) $N_{SPOR}$ = number of sporangia in the central 1000 × 1000 pixels, (3) $A_{SPOR}$ = area of the largest 10 sporangia, (4) $A_{ZOOSPORE}$ = area of 10 random

spores. To measure fecundity, at day 7, the following formula was used ((average $N_{ZOOSPORE}$/average $N_{SPOR}$)/average $A_{SPOR}$) as described in Fisher et al.[26].

The number of zoospores and the number of sporangia were modelled using generalized linear models (GLM) with Poisson error distributions, while the area of sporangia and the calculated fecundity were modelled using linear models, all with isolate as the response variable. For the area of sporangia, a random effect at the replicate level was included to account for pseudo-replication. The significance of pairwise differences among isolates was assessed using Tukey's HSD test. In addition to differences among isolates for individual characteristics, their overall similarity was assessed using two-dimensional non-metric multidimensional scaling (NMDS). NMDS maps the relationship between the dissimilarity matrix (Bray–Curtis index) to locate each sample along two coordinates in the ordination space[51]. All analyses were performed in R[52].

**Infection trials.** The animal experiments were performed with the approval of the ethical committee of the Faculty of Veterinary Medicine (Ghent University EC2016/20, EC2015/86). Only captive-bred animals were used in standardized experiments, using identical environmental conditions that allow comparison of isolate virulence and host susceptibility. All animals were housed individually in terraria at 15 °C on moist tissue with access to a hiding place. All animals (males/females) were clinically healthy and derived from breeding colonies that are free of *Bd, Bsal* and Ranavirus as assessed by sampling the skin using cotton-tipped swabs and subsequent performing qPCR or PCR[46,47,53]. Individuals were randomly assigned to treatments. All animals were clinically inspected daily. Skin sampling was done weekly and the swabs were analysed for the presence of *Bd* using qPCR described by Hyatt et al.[53] with the respective isolate used as standard. Sample analysis was blinded. Animals were euthanized by an overdose of pentobarbital. Humane endpoints were set at the loss of self-righting ability and/or change in posture.

**Susceptibility of native amphibians to *Bd*GPL isolates under standardized laboratory conditions.** Five captive-bred individuals of seven species (fire salamander (*Salamandra salamandra*), common spadefoot (*Pelobates fuscus*), natterjack toad (*Epidalea calamita*), common toad (*Bufo bufo*), alpine newt (*Icthyosaura alpestris*), great crested newt (*Triturus cristatus*) and European tree frog (*Hyla arborea*)) were exposed to the local *Bd*GPL isolate *Bd*BE1 or the hypervirulent isolate *Bd*JEL423. In addition, two midwife toads (*A. obstetricans*) were included as *Bd*JEL423-infected positive control group (the limited number of midwife toads in this experiment were merely included as positive controls, extensive experiments in this species were conducted separately). For inoculum, isolates were grown on TGhL agar at 20 °C, for 5 days. Plates were washed with sterile distilled water, and the water containing the zoospores was passed over a sterile mesh filter with pore size 10 mm (Pluristrainer, PluriSelect), then counted using a haemocytometer and inverted microscope (Nikon Eclipse TS100, Nikon). Spores were resuspended at a concentration of $1 \times 10^6$ spore ml$^{-1}$. Individuals were inoculated with 1 ml of $1 \times 10^6$ fresh spores, in separate Petri dishes and placed in climate-controlled (~15 °C & ~80% relative humidity) room for 24 h. Individuals were then transferred to individual plastic containers containing damp tissue and a hide. Small crickets were given ad libitum, providing a constant food supply. Clinical examination of the animals was carried out daily, and tissue replacement and swabbing weekly. Species that did not show infection for over 3 weeks (3 weeks negative in qPCR and no clinical signs) in any individual were considered uninfected and removed from experiment at week 8. Absence of *Bd* infection was confirmed by histopathology and immunohistochemistry. Sample analysis was blinded. We compared the proportion of individuals of each species becoming infected from either isolate using a generalized linear model with binomial error distribution.

**Differential *Bd*GPL isolate pathogenicity for midwife toads.** Four local phenotypically and genotypically[1] different isolates originating from different species (Fig. 1, Supplementary Table 5) (*Bd*BE1, *Bd*BE3, *Bd*BE4 and *Bd*BE5) and the hypervirulent isolate *Bd*JEL423 were selected to inoculate midwife toads (*A. obstetricans)*. This species is used as model species for susceptibility to virulent *Bd*. Negative control animals were sham inoculated with water. Thirty-six newly metamorphosed *A. obstetricans* were randomly assigned to groups of six for each treatment. Individuals were exposed to 1 ml of $1 \times 10^6$ of fresh spores, or distilled water (prepared as previously) for 24 h. Individuals were then transferred to individual plastic containers containing damp tissue and a hiding place in a climate-controlled (~15 °C & ~80% relative humidity) room. Small crickets were given ad libitum, providing a constant food supply. Clinical examination of the animals was carried out daily, and tissue replacement and swabbing weekly. From day 15 onwards, animals were weighed weekly. Individuals that died during the trial or were euthanized due to reaching a humane end point (see above) had skin samples taken for histology and DNA extraction for qPCR. Since after the first inoculation none of the toadlets became positive for *Bd*BE5, in week 5, an additional five *Bd*BE5 and three *Bd*JEL423 individuals were inoculated and added to the experiment. Sample analysis was blinded.

The probability of an individual dying as a result of infection with different isolates was modelled using a binomial GLM. Penalized regression (package *brglm2* in R[54]) was used to account for perfect separation in some groups, i.e. where all or

no animals died within a given treatment group. Since death occurred for all *Bd*JEL423-infected animals and for only one of the animals infected with the local isolates, parametric survival regression model could not be fitted to estimate the mortality rate over time. The relationship between probability of death and the infection load sustained was modelled using a binomial penalized GLM (as above). Infection load was always modelled as the $\log_{10}$ of the genomic equivalent. Mean, maximum and total load over the course of infection were all assessed as predictors.

**Protective capacity of low-virulence *Bd*GPL against hypervirulent *Bd*GPL challenge in midwife toads.** After completion of the multi-isolate *A. obstetricans* infection trial, all remaining individuals were exposed to 1 ml of $1 \times 10^6$ of fresh spores of *Bd*JEL423 (prepared as previously) for 24 h. Individuals were then transferred to individual plastic containers containing damp tissue and a hiding place in a climate-controlled (~15°C & ~80% relative humidity) room. Small crickets were given ad libitum, providing a constant food supply. Clinical examination of the animals was carried out daily, and tissue replacement and swabbing weekly. Sample analysis was blinded.

As for the initial infection trials, penalized binomial GLMs were used to estimate whether the probability of death following infection with *Bd*JEL423 depended on (1) the isolate individuals were exposed to in the initial experiment and (2) the mean/maximum/total infection load sustained during the initial experiment. Parametric survival regression was also used to assess whether the survival rates of *Bd*JEL423-re-infected individuals depended on (1) the isolate they had been exposed to in the initial trial and (2) on the mean load of infection in the initial trial. In both cases, the response variable (death) was modelled using a Weibull distribution while accounting for censoring. We repeated all analysed grouping individuals initially infected with local isolates *Bd*BE1/*Bd*BE3 and *Bd*BE4/*Bd*BE5, since these two pairs of isolates exhibited similar characteristics in previous experiments.

To confirm the protective capacity of *Bd*BE3, 40 newly metamorphosed *A. obstetricans* were randomly assigned to two groups and exposed to 1 ml of $1 \times 10^6$ of fresh spores of *Bd*BE3, or distilled water (prepared as previously) for 24 h. Animals were then housed individually in plastic containers containing tissue, a hiding place and water dish in a climate-controlled (~20 °C & ~80% relative humidity) room and examined as previously described. Four weeks later, all animals were exposed to 1 ml of $1 \times 10^6$ of fresh spores of *Bd*JEL423 (prepared as previously) for 24 h. Individuals were again transferred to individual plastic containers containing tissue, a hiding place and water dish. Small crickets were given ad libitum, providing a constant food supply. Clinical examination of the animals was carried out daily, and tissue replacement and swabbing weekly. Sample analysis was blinded. The readout for this experiment was the comparison of the proportion of *Bd* infected animals after challenge with the highly virulent *Bd*JEL423 in each of the two treatment groups. Significance of protection was calculated using two-tailed Fisher's exact test.

**_Bd_ growth in amphibian mucosomes.** Mucosomes were collected from healthy midwife toads (*Alytes obstetricans*) and fire salamander (*Salamandra salamandra*) using the bathing method described by Woodhams et al.[55]. For each species, mucosomes were collected from ten animals (animals were not used in the previous experiments). Briefly, each animal was bathed in a Petri dish with HPLC water for 1 h, the volume of water was calculated according to animal surface[56,57]. The collected mucosome solutions were filtered through a 0.2 μm unit filter (Whatman, GE Healthcare Life Sciences) and immediately used in the next steps. Two *Bd*GPL isolates, the hypervirulent *Bd*JEL423 and the local *Bd*BE1 were used in this experiment. For each isolate, spores were collected from culture flasks with sporulating sporangia. The medium was collected and filtered using a sterile filter with pore size 10 μm (Pluristrainer, PluriSelect). To achieve the target concentration of $1 \times 10^6$ spores per mL, the spore suspension was diluted with TGhL broth. Finally, 100 μl of this spore suspension was used to obtain $10^5$ spores per well of a 96-well flat-bottom plate (Greiner Bio-One).

In each well, 100 μL of the mucosome solution was added to $10^5$ spores. As positive control, sterile water was added instead of mucosome and the negative control consisted of heat-killed spores (10 min at 100 °C). All the samples were performed in triplicate. After incubation at 20 °C for 5 days, the bottom of each well was scraped with a 100 μL tip and the liquid was transferred to a 1.5 ml Eppendorf tube for DNA extraction and qPCR[58]. Growth of *Bd*BE1 and *Bd*JEL423 (corrected GE values) was shown by subtracting the mean GE loads of the negative control from the original GE values. Multiple comparisons on the original values were assessed by a Kruskal–Wallis analysis, followed by pairwise Mann–Whitney U-test with a Bonferroni-corrected *P* value of 0.017 (*P* value = 0.05/3 pairwise comparisons) (SPSS version 26).

**_Bd_GPL gene expression.** We compared gene expression of a selection of virulence genes as identified by Farrer et al.[5]. Using the RNeasy mini kit (Qiagen), total RNA was isolated from fresh *Bd* spores ($2.5 \times 10^7$ spores per condition) and *Bd* spores ($2.5 \times 10^7$ spores per condition) that were incubated for 2 h at 20 °C with chytrid negative skin of *A. obstetricans* collected with a skin biopsy punch (6 mm). *Bd* spores were collected by putting sterile distilled water on a culture flask containing mature sporangia. Once the zoospores were released, the water containing the

zoospores was collected. In order to reduce the percentage of mature cells, the water containing the zoospores was passed over a sterile mesh filter with pore size 10 μm (Pluristrainer, PluriSelect). The flow-through was used as the zoospore fraction (>90% purity). All conditions were analysed with *Bd* spores originating from *Bd*JEL423, *Bd*BE1-*Bd*BE10 and they were tested in fourfold (biological replicates *n* = 4). Total RNA (1 μg) was reverse transcribed to cDNA with the iScript cDNA synthesis kit (Bio-Rad). The housekeeping genes α-centractin, APRT, TUB and Ctsyn1 were included as reference genes[59]. The list of genes and sequences of the primers used for quantitative PCR analysis can be found in Supplementary Table 10. Real-time quantitative PCR reactions were run in triplicate (technical replicates *n* = 3) and the reactions were performed in 10 μl volumes using the iQ SYBR Green Supermix (Bio-Rad). The experimental protocol for PCR (40 cycles) was performed on a CFX384 RT-PCR cycler (Bio-Rad) and data were analysed using the Bio-Rad CFX manager 3.1. The results are shown as fold changes of mRNA expression relative to the mRNA expression levels in fresh spores. Fold changes were calculated using the cycle threshold ($\Delta\Delta C_T$) method, and they were analysed in SPSS version 25 (SPSS Inc., Chicago, IL, USA). Multiple comparisons were assessed by a Kruskal–Wallis analysis, followed by pairwise Mann–Whitney U-tests adjusted for multiple testing with a Benjamini–Hochberg correction[60], setting an adjusted *P* value of 0.05 as significant. Correlation between the relative expression of each isolate to *Bd*JEL423 for CRN-like genes of fresh spores exposed to midwife toad skin and colonisation capacity from the individuals from the multi-isolate *A. obstetricans* infection trial was assessed by Spearman's rank correlation in R[54].

**In vitro infection of A6 cells**. Here, we compared the invasive capacity between hypervirulent and low-virulence local *Bd*GPL isolates using a cell culture model. The *Xenopus laevis* kidney epithelial cell line A6 (ATCC-*CCL 102*) was grown in 75 cm² cell culture flasks and maintained in complete growth medium (74% NCTC 109 medium, 15% distilled water, 10% fetal bovine serum (FBS) and 1% of a 10,000 U ml⁻¹ penicillin-streptomycin solution (P/S)) and the cells were incubated at 26 °C and 5% $CO_2$ until they reached confluence. Using trypsin, the cells were detached, washed with 70% Hanks' Balanced Salt Solution without $Ca^{2+}$, $Mg^{2+}$ (HBSS−) by centrifugation for 5 min at 1500 rpm and resuspended in the appropriate cell culture medium for invasion assays, which were performed as described in Verbrugghe et al.[61]. To assess the germ tube formation, A6 cells were stained with 3 μM CellTracker™ Green CMFDA, seeded (10⁵ cells per well) in 24-well tissue culture plates containing collagen-coated glass coverslips and they were allowed to attach for 2 h at 20 °C and 5% $CO_2$. After washing three times with 70% HBSS+, they were inoculated with *Bd* zoospores in invasion medium, at a MOI of 1:10. Two hours p.i., the cells were washed three times with 70% HBSS+ and the invasion medium was replaced by staining medium. Four hours p.i., the infected cells were washed three times with HBSS+ and they were incubated with Calcofluor White stain (1 μg ml⁻¹ in 70% HBSS+) for 10 min. After washing three times with 70% HBSS+, the cells were fixed, mounted and analysed using fluorescence microscopy. To assess the invasive growth, A6 cells were seeded and inoculated with *Bd* zoospores as described above. Two days p.i., the infected cells were stained with 3 μM CellTracker™ Green CMFDA, washed three times with 70% HBSS+ and they were incubated with Calcofluor White stain (10 μg ml⁻¹ in 70% HBSS+) for 10 min. After washing three times with HBSS+, the infected cells were fixed, permeabilized for 2 min with 0.1% triton and incubated for 60 min with a polyclonal antibody against *Bd* (1/1000), which was obtained by immunizing rabbits with *Bd*-antigen[62]. After washing three times with 70% HBSS+, the samples were incubated with a goat anti-rabbit Alexa Fluor 568 (1/500). After an incubation of 1 h, the samples were washed three times with 70% HBSS+, mounted and analysed using fluorescence microscopy and Leica Application Suite (LAS) software X. The Alexa Fluor 568 targeting *Bd* and Calcofluor White stainings are used in concert to assess the ability of *Bd* to penetrate the host cell. Calcofluor White is not internalized by A6 cells, whereas the Alexa Fluor 568 targeting *Bd* staining was applied after permeabilisation of the host cells. As such, intracellular *Bd* will only be targeted by the Alexa Fluor 568 and extracellular *Bd* bodies will be bound with both the Alexa Fluor 568 and Calcofluor White stain. Overlay pictures were made with ImageJ 1.52d software. To assess the in vitro infection dynamics of different *Bd*GPL isolates, three independent in vitro experiments were conducted with every condition being tested in triplicate, with similar results.

**Whole-genome sequence analysis**. WGS read data were downloaded from NCBI Bioproject PRJNA413876 (SRA sample accession numbers: *Bd*BE1; SRA: SRS2757215, *Bd*BE3; SRA: SRS2757203, *Bd*BE4; SRA: SRS2757202, *Bd*BE5; SRA: SRS2757217, *Bd*JEL423; SRA: SRS2757141). Reads were aligned to the reference *Bd*JEL423 assembly (BioProject PRJNA13653) using BWA mem version 0.7.17[63] and the aligned bam files were processed using Picard tools version 2.21.1 (http://picard.sourceforge.net/) AddOrReplace, MarkDuplicates, SortSam, CreateSequenceDictionary and ReorderSam. Variants were called using GATK v4.1.4.0[64] HaplotypeCaller, the output GVCFs were combined using CombineGVCFs, genotyped using GenotypeGVCFs, separated into SNP and Indel variants for filtering using SelectVariants and filtered using VariantFiltration with filters QD < 2.0 || FS > 60.0 || MQ < 40.0 || MQRankSum < −12.5 || ReadPosRankSum < −8.0 for SNPs and QD < 2.0 || FS > 200.0 || ReadPosRankSum < −20.0 for Indels. Variants were annotated using SNPeff v4.3[65] and the *Bd*JEL423 reference protein annotation

(BioProject PRJNA13653), with those variants annotated as HIGH impact on function used for further analysis. Variants that were only present in all local isolates or only present in the *Bd*JEL423 isolate were identified as the genes of interest for the virulence/local phenotype. Similarly, variants that were only present in *Bd*Be1, *Bd*Be3 and *Bd*JEL423 while absent in *Bd*Be4 and *Bd*Be5, or variants present in *Bd*Be4 and *Bd*Be5 while absent in *Bd*Be1, *Bd*Be3 and *Bd*JEL423 were identified as variants of interest for the invasive/epibiotic phenotype. The *Bd*JEL423 reference protein annotation was further annotated using SignalP 4.0 server[66], the TMHMM Server 2.0[67] and the HMMER 3.2.1 server[68] using the gathering threshold. Candidate Crinkler proteins were identified using Blastp (BLAST+ 2.9.0)[69], all candidates listed displayed 100% identity and e-value of 0. PFAM domain enrichment in variant-effected groups was identified with right-tailed Fisher's exact test using *q*-value False Discovery Rate. Copy number variation comparisons were made using CNVkit[70] version 0.9.5 batch, segment, scatter, heatmap (with –d flag to de-emphasize low-amplitude segments) and genemetrics commands, utilising the JEL423 reference assembly and annotation, specifying the *Bd*JEL423 read aligned bam as the normal control compared to the local isolate samples.

**Protective capacity of low-virulence *Bd*GPL against highly virulent *Bsal* in three urodele species**. Clinically healthy, *Bd*, *Bsal* and ranavirus free fire salamanders (27), ribbed newts (*Pleurodeles waltl*) (24) and marbled newts (*Triturus marmoratus*) (20) were divided ad random in three treatment groups (every animal was housed individually. Two groups of each species were exposed to local *Bd*GPL (1 ml of *Bd*SP11 10⁶ spores for 24 h), 3 weeks later one of these groups to *Bsal* (1 ml of AMFP18/02 10³ spores for 24 h). At this 21-day time point, a *Bsal* control group of each species was exposed to *Bsal* (1 ml of AMFP18/02 103 spores for 24 h). All animals were clinically inspected daily. Skin sampling was done weekly and the swabs were analysed for the presence of *Bsal* using qPCR as described by Blooi et al.[46,47] with the respective isolate used as standard. *Bsal* infection was confirmed using histopathology. Sample analysis was blinded. Statistical analyses were carried out using generalized linear models and survival regression, following methods described for *Bd* infection trials above (with treatment group in place of isolate).

**Reporting summary**. Further information on research design is available in the Nature Research Reporting Summary linked to this article.

## Data availability

All data supporting this study are available in the article and corresponding supplementary information files. Genomic read data are available at NCBI GenBank under Bioproject PRJNA413876 (SRA sample accession numbers: BdBE1; SRA: SRS2757215, BdBE3; SRA: SRS2757203, BdBE4; SRA: SRS2757202, BdBE5; SRA: SRS2757217, BdJEL423; SRA: SRS2757141). Reads were aligned to the BdJEL423 assembly available under BioProject PRJNA13653, assembly accession GCA_000149865.1. Source data are provided with this paper.

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

## Acknowledgements

This work was funded by the Ghent University Special Research Fund (BOF16-GOA-024.08) and the 'Agentschap voor Natuur en Bos' (BL-100/FIN/15-00229). E.V., M.K. and W.B. were supported by the Research Foundation Flanders (FWO grants 12E6616N, 1507119N, 1111119N, 11ZK916N-18N). Z.L. is supported by the China Scholarship Council (CSC, 201707650012) and Ghent University Special Research Fund (BOF17/CHN/018). We thank Iwan Lewylle for the field-work assistance. Kim Roelants kindly provided the artwork.

## Author contributions

A.M. and F.P. designed the research. A.M., M.S.G., E.V., M.K., M.B., W.B., N.D.T., Z.L., G.S., R.V.L. and S.V.P. carried out the research. A.M., M.S.G., E.V., M.K., W.B., S. Canessa, L.L., Z.L., D.S., R.V.L. and F.P. analysed the data. M.S.G., E.V., W.B., S. Canessa, S. Carranza, S.C., D.F.G., P.G., L.L., Z.L., D.S., R.V.L., M.V.E., M.V., F.P. and A.M. interpreted the data. A.M., F.P., M.S.G., E.V., M.K., W.B. and S. Canessa wrote the paper with input from all other authors.

## Competing interests

The authors declare no competing interests.
