## [Peer Review File · Nature Communications]

Reviewers' Comments:

Reviewer #1:

Remarks to the Author:

Greener et al. explore the effects and interactions of low and virulent isolates of *Batrachochytrium dendrobatidis* (Bd) and *B. salamandrivorans* (Bsal) on amphibians. The study addresses an important question, but unfortunately, this is a classic case where the authors have chosen breadth at the expense of depth and rigor of experimental design. The manuscript includes field sampling (n=5), phenotypic characterization of multiple Bd isolates, infection trials of amphibians (n=3-6), tests of the protective capacity of low virulent Bd against high virulent Bd (n=6), Bd growth in amphibian mucosome (n=3), gene expression trials of Bd, in vitro infection of A6 cells (I have no idea what the sample size was because I don't believe it was provided), whole genome sequence analysis (unknown sample sizes because never provided), and trials on the protective capacity of Bd to Bsal in salamanders (n=6-9). Most of the sample sizes are either not provided or between 3 and 6 for the most salient studies on interactions between the fungi and the amphibians. Most of the statistical analyses are not significant because these sample sizes are highly insufficient for quantifying Bd or Bsal loads because these response variables have negative binomial error distributions. The rule of thumb for negative binomial error distributions is that 80% of the pathogen is found on 20% of the individuals, requiring much larger sample sizes than the authors chose to properly characterize responses. Moreover, the authors ignore the amphibians that did not get infected, probably to reduce variability and increase statistical power, but these resistant amphibians should not be ignored. This could greatly alter the conclusions. Sample sizes between 3 and 6 are even more insufficient for detecting effects on mortality, which is probably why most effects on mortality were not significant (possibly false negatives?). Finally, the studies on the effects of low virulent Bd on higher virulent fungi confounded selection and phenotypic plasticity because the authors let considerable mortality occur from the first exposure to the fungi. I also found it surprising that the manuscript ignored studies and theory on coinfections, given that this seemed to be a major focus of the current work.

In general, I found that the studies lacked necessary rigor to support the conclusions drawn by the authors, that the authors speculated too much in the Discussion, and that the authors did not acknowledge the many assumptions and caveats necessary to support their conclusions (e.g., no or weak host-by-parasite genotype interactions, no drift due to low sample sizes). I do think there is a lot of valuable information in this manuscript, but for all the reasons just described, I cannot support publishing this work unless the authors can increase their sample sizes, reduce speculation, and better acknowledge the many important assumptions associated with their conclusions.

L 60: Your study doesn't quantify the costs of endemism. This was just a survey of Bd on amphibians. Please revise accordingly. Speculation.

L 92-100: Sample sizes are way too small. Concluding that midwives toads are sentinel species is ridiculous. You have tested only two isolates of Bd and never provide a definition of a sentinel species. This also assumes no host genotype by parasite genotype interactions or at least the interaction is weak relative to the main effects. All of this is being concluded with a sample of five individuals per species, which is way too low given that Bd is distributed in a negative binomial manner (i.e., where most of the parasites are found on a small number of hosts)! Several studies suggest that sample sizes of 15 or higher are needed to properly capture the distributional patterns of many negative binomially distributed pathogens.

L 105-109: But, if these data were collected from the same individuals, the trials are not independent and thus are simply autocorrelated. Was that the case?

L 121-138: Based on what sample sizes? I could not find this in the Methods.

L 136-138: I don't understand these statistics? What do CRN_#s mean? CRN is never defined in the manuscript. Also, what are the sample sizes? They are never provided.

L 171-174: Nothing is significant because a sample size of six toads is way too low.

L 177-187: These studies confound selection with plasticity because the infections were allowed to induce significant mortality.

L 183-185: Again, nothing is significant because the sample sizes are insufficient to detect effects on mortality.

L 194-196: Not sufficient sample sizes to conclude this.

L 200-203: So perhaps this is not evidence of Bd genotype changes but evidence of host species variation?

L 204-209: I don't agree with any of this because the authors have completely ignored the possibility of g x g interactions. Perhaps state that, assuming no parasite genotype-by-host genotype interactions...

L 212-215: By stating "would depend", this is stated as if your study was causal. Your study was only correlational. You don't know that these differences were the cause of the different susceptibility.

L 218-220: Unjustified speculation.

L 221-224: These studies confound selection with acquired resistance and thus cannot say anything about acquired resistance at all. They were not done in a manner where the infections were cleared before substantial mortality and thus we do not know whether any susceptibility differences were due to selection or plasticity. Another example of where the authors attempt to do too much in my opinion in a single study without doing many of the experiments well (insufficient sample sizes, rigor, experimental design, and controls).

L 224-227: Again, way too much speculation. Conduct the experiments or develop the mathematical models rather than excessively speculate on how these dynamics will play out.

L 228-229: Not an explanation, a hypothesis! Please be more precise with your language. You haven't explained these extirpations at all. You simply offer a hypothesis.

L 232-234: I don't find this result convincing at all for two reasons. First, insufficient sample sizes and second, it is unclear whether the results are due to selection or plasticity.

L 234-236: This study did not test the hypothesis that this increases risk to highly susceptible hosts. This should be stated as a hypothesis that deserves testing. Please make this clearer.

L 236-238: Perhaps, but this assumes that either 1) there is no host genotype-by-parasite genotype interaction or 2) that this interaction accounts for much less variation than the main effect of parasite genotype. We have no evidence to support either of these assumptions to which I am aware. Thus, I cannot advocate this overall conclusion.

L 334: Five individuals are not enough to draw conclusions about the population given that loads are distributed as negative binomial error distributions, i.e., most of the zoospores are found on a very small number of individuals. And, five individuals will unquestionably be insufficient to detect effects on mortality.

L 352-354: Why are you ignoring individuals that did not become infected? These individuals are important in assessing resistance of the population. This is a big issue for me.

L 357: Too many adjectives strung together.

L 361-362: Six animals is not enough per treatment group to assess time of death. Sample sizes are way too low.

L 365: What is "a hide"?

L 368: What were the humane endpoints that induced euthanasia? Your study needs to be replicable.

L 369-371: Huh? This is a really weird change in methods that makes the results difficult to interpret. Again, a sample size of 5 and 3 is simply insufficient for time of death trials.

L 381-383: The problem with this approach is that you have induced a selection event given that there was considerable initial death. Given that the sample sizes are extremely small, any initial death, whether it be due to treatment or stochastic, will increase the chances of drift as well. This is all too problematic for me to feel confident in any of the conclusions.

L 394-395: A Weibull distribution does not account for censoring. Please revise to improve accuracy.

L 395-397: You did this because you don't have adequate statistical power because your sample sizes

are too low across all experiments.

L 398-400: You are trying to do way too much in this study. As a consequence, you have done nothing well. Sample sizes are simply too low.

L 400-401: A sample size of three for the mucosome experiment is completely insufficient.

L 447-478: What are the sample sizes for the in vitro infection of A6 cells? I can't find it documented anywhere.

L 508-511: Sample sizes are 6-9, which is better than the other studies, but still insufficient for detecting effects on mortality, especially for cases with 6 animals.

L 511-515: Why these treatments in these orders?

Figure 1: How is fecundity different than number of zoospores? Please define here given that figures and their legends are supposed to be stand alone.

Figure 3: Cool, but there are just too few toads. The sample sizes are too low. You are comparing 5 deaths to 1 or 2 deaths. Chance alone can explain these results. This is why nothing was significant in the Results section.

Reviewer #2:

Remarks to the Author:

What are the major claims of the paper?

In this manuscript, Greener et al. describe a study that investigated infectious disease dynamics of fungal pathogens of amphibians. The investigators integrated field surveys and multiple experimental investigations in an attempt to understand the importance of pathogen endemism in determining disease dynamics in a specific amphibian host assemblage. Specifically, they aim to address the question of why or how the infectious disease chytridiomycosis has had different impacts on amphibians around the world.

The authors' primary claims are that 1) the fungal pathogen Bd is co-existing with amphibian hosts in Europe (Belgium), with no strong evidence of recent disease-induced declines, and 2) the presence of low-virulence Bd (in this case, Bd isolates that originated in Europe, which they term "endemic") may protect, or provide a barrier, for amphibians against more virulent Bd (which they term "exotic") invasion.

While these central claims are intriguing, I find that they are an odd fit for an otherwise highly compelling study. As I detail in my review below, I believe many aspects of this study are remarkable in terms of their integration, rigor, and scope to understand the pathophysiology of chytridiomycosis and the differential pathogenicity of Bd isolates. On this count, the authors provide numerous insightful and novel findings that will advance our understanding of this disease and be of interest to a broader audience. However, I am unsure if the strong points of this study provide robust support for their central claims as outlined in the paper. Below I provide some additional detail on why their claims regarding Bd endemism may benefit from additional work and/or clarification.

Is the work convincing?

To reiterate, I find this study to be convincing and exciting for multiple reasons, but they are reasons that differ from those that the authors have articulated. To begin with, I believe they have captured the molecular and cellular pathophysiology of chytridiomycosis in a way that is novel and highly effective. While previous studies have used similar experimental designs in a rather piecemeal fashion, this study integrates multiple innovative approaches in an elegant way and has applied them to

multiple host species. The insights provided on this topic will be of great interest to a diverse community of researchers. In addition, I believe this study provides a detailed mechanistic explanation for differential virulence among Bd isolates. The experiments were very well executed and I find them to be highly convincing. These aspects of the study address long-standing questions that are critical to resolving the devastating impacts of this disease. These accomplishments are to be lauded and I expect they will significantly advance this field of study.

With that said, I am less enthusiastic about some of their central assertions regarding 1) why host assemblages differ in their responses to Bd emergence and 2) that infection with a low-virulence isolate provides a barrier to invasion with a more virulent isolate. In fact, I would strongly discourage the inclusion of several of their inferences for reasons that I outline below.

If not, what further evidence would be required to strengthen the conclusions?

1) The authors observed different disease outcomes following exposure to multiple Bd isolates (e.g., Fig 3). They attribute these findings to low-pathogenicity due to isolate endemism (endemic European isolates) and suggest that this may be why impacts of the disease vary in Europe (lines 48-50 and 189-191). Their rationale is primarily based on their finding that the rates of mortality were lower in European amphibian hosts when exposed to European Bd isolates compared an isolate from Central America. While I agree that they have successfully documented differential pathogenicity, it is unclear to me that this is due to the geographic point of origin of the isolates (i.e., isolate endemism) and/or that the results can explain the variable disease impact in Europe. Here are two additional pieces of information that should be considered and additional suggestions that may bolster this particular argument:

- Pathogenicity can shift in vitro. We know that pathogen virulence (including for Bd) is malleable, dynamic, and can change depending on a wide range of factors, including isolation and passage methods, media, incubation temperatures, etc. As such, it would be helpful to know if all of the isolates used in this study have always had the same (or comparable) in vitro maintenance. Furthermore, there are methods to mitigate the effect of differences in pathogen maintenance. For example, using an approach where all of the isolates are cryopreserved, revived, and subsequently passaged identically in the laboratory could address this issue. If the authors considered this question, and used methods to address this prospective problem, they are not currently presented in the paper. Without this information, it is reasonable to consider that the results were due to treatment of the isolates prior to the experimental work, rather than the geographic point of isolation.

- Differential pathogenicity, not endemism, may be the most parsimonious explanation. We have known for many years that different Bd isolates, even from within the same strain/lineage, can have variable disease outcomes. As such, it is possible that the authors have documented different levels of pathogenicity that are unrelated to the geographic origins of the isolates. In other words, it may not be appropriate to explain their results through the lens of exotic versus endemic isolate pathogenicity. One way to address this question would be to include parallel group of "exotic" host amphibians in their exposure experiments (e.g., using a non-European, neotropical host species). This approach would provide a fully factorial study design that more directly examines "exotic versus endemic" patterns. If an exotic (Central American) species had been similarly exposed to all isolates, the results might have revealed an opposite pattern, where the exotic hosts exhibited high mortality when exposed to the European isolates (* but see note below). Such a result would support their central claims. In addition, they could have included additional "exotic" Bd isolates to provide biological replication. Both of these modifications to the experimental design might have provided more direct evidence to support their argument that the European isolates were less pathogenic due their endemism (and therefore help explain disease dynamics in Europe).

*As an additional note, my suspicion is that this particular "exotic" isolate is also highly pathogenic to amphibian species in Central America. There is evidence for this in the literature (DiRenzo et al. 2014, Ellison et al. 2014a and 2014b, Voyles et al. 2018) where this isolate has caused up to 100% mortality in Central American and North American amphibian host species.

Without this additional information, it is not clear that we can rule out the possibility that JEL423 is simply a more pathogenic isolate or has become one over the course of laboratory maintenance. While we may be able to conclude that the European isolates are less pathogenic (at this point in time) by comparison, it is not clear to me that this finding is wholly due to the fact that they come from Europe. They may simply be inherently less pathogenic.

Taking all this into account, I suggest that this conclusion statement may require some additional consideration and adjustment:

"Although Bd has caused death and species extinction on a global scale, its impact in Europe is limited. This can be explained by the fact that a limited number of European amphibian species are susceptible to Bd-induced chytridiomycosis, and by the presence of low virulence endemic BdGPL lineages that can protect susceptible species against virulent exotic BdGPL." (Lines 189-192)

Not only is the importance of isolate endemism questionable (for the reasons explained above), but this explanation also fails to acknowledge the many spatio-temporal factors that strongly influence disease impacts (e.g., time since pathogen emergence, local and regional ecological factors, community composition, etc.).

Additional comments-

It is possible that addressing the points outlined above could alleviate several issues. However, I wanted to point out some additional concerns with the following terms: "endemic" and "exotic" Bd and "lineage" or "strain" vs. "isolate"

I suggest that the approach that the investigators used is essentially a comparison of multiple isolates from within the same BdGPL clade (lines 82-83). As such, it is probably inappropriate to use terms such as "BdGPL lineages" (lines 191-192) as it misrepresents the fact that they are all BdGPL isolates. This comment raises a slightly larger issue regarding the merits of comparing pathogenicity among the European isolates (e.g., BdBE1, BdBE3, BdBE4, BdBE5; Fig. 3). Because these isolates appear to be the same in terms of genetics (and most phenotypic characteristics), one could argue that the exposure experiment reflected a form of pseudoreplication (i.e., multiple experimental groups with the same genetic strain). However, I am not making that argument because 1) I noted that these isolates come from different amphibian host species and/or life stages (Supplementary Table 4), and 2) the strength of their pathophysiology findings are a result of the documented variation among isolates (possibly reflecting a subset of their phenotypic characteristics, such as number of sporangia; Fig. 1). Because these are not different Bd lineages, but rather multiple isolates of the same strain, perhaps a more accurate explanation is that the isolates used in this experiment were the most different in terms of phenotype and/or host origin/life stage. This terminology would be preferable to describing the isolates as different lineages.

2) The investigators suggest that infection with a low-virulence isolate provides a barrier to invasion with a more virulent isolate. Their support for this claim comes from the sequential exposure of toads to the low pathogenicity isolates (the European isolates) and subsequently to the high pathogenicity isolate (the Central American isolate) as well as a similarly designed experiment with salamanders. I have three concerns regarding part of the experiment:

- Regarding the methods and experimental design, it is unclear to me if the investigators controlled for the infection load at the time of the exposure to JEL423? I imagine that, with this study design, the investigators may have created a scenario for super infection (i.e., within host competition between two isolates). As such, the infection intensities may reflect the success of one or the other of the isolates. In order to evaluate this possibility, it would have been helpful to include parallel groups that received secondary exposures to the European isolates only (as a positive control) or if there was a way to distinguish between the isolates using a discriminatory PCR approach.

- Upon evaluation of the data provided in Supplementary Table 8, it appears to me that only one group of toads exposed to a low-virulence isolate had a significantly higher survival rate (BdBE3). This result concerns not only because the sample sizes are small, but also because it appears that only 1/4 of the groups with prior exposure to a low-virulence isolate had increased survival. I also considered the results from the exposure of salamanders to Bd and subsequently to Bsal, which were inconsistent across experimental groups. To me, the take-away message from these experiments is that the results of prior infection with a low-virulence isolate were mixed.

- The claim that a low-virulence isolate provides a "barrier" to subsequent does not appear to be fully supported. The subsequent exposures resulted in infection, and the maintenance of infection over time, rather than a resistance to infection. So, while it is not clear to me how infection operated as a barrier, perhaps the authors have a different interpretation of this term? If so, the use of this term may require some additional explanation.

In raising these concerns, I would like to note that the accurate interpretation/presentation of these results will be highly consequential for conservation and management practices. To be sure, understanding the dynamics underpinning co-infection is incredibly interesting and exciting for academic reasons. However, I urge caution, especially with suggestions of "natural vaccination" (lines 233-234), for several reasons: we still have a limited understanding of co-infection processes, pathogen virulence can shift, and the repercussions of misperceptions in management scenarios could be grave. For example, the authors suggest that infected populations "might be protected against disease" (in lines 210-212). This statement is both speculative and alarming because, taken out of context, it could be used for a wide range of conservation actions (or inaction) that could be detrimental to amphibians based on our understanding of chytridiomycosis globally.

Do you feel that the paper will influence thinking in the field?

Potentially. While I am very enthusiastic about the core scientific content of their work, I am less certain of the "packaging" (i.e., their explanations and central claims). I believe they have not showcased their results in a way that fully captures how unique and exciting their study is. Specifically, they uncover critical mechanistic processes of a notorious disease system. Their findings are revelatory for the community of researchers working on chytridiomycosis as well as a broader audience interest in emerging infectious disease.

Minor comments-

Throughout the paper- The term "pathogenicity" or "pathogen virulence" may be preferable because in many fields, the term "virulence" is a composite trait, involving the host and the pathogen. Please see comments above regarding terms such as "lineage".

Lines 25-28 The opening sentence is rather long and the subject of the sentence is narrowly focused.

Might it be better to use a sentence that appeals to the broader scientific community?

Line 28 Is "populations" the best term? I suggest "communities" or "assemblages".

Line 30 and 61-62 The term "widespread" is pretty subjective and seems odd given that the results seem to be spatially limited to Belgium.

Lines 45-46 I believe this sentence does not reflect some well documented Bd-induced declines in North America (e.g., Briggs et al. 2010), Europe (e.g., Bosch et al. 2001), and Africa (e.g., Weldon et al. 2019)

References

Bosch, J., Martínez-Solano, I., & García-París, M. (2001). Evidence of a chytrid fungus infection involved in the decline of the common midwife toad (*Alytes obstetricans*) in protected areas of central Spain. *Biological conservation*, 97(3), 331-337.

Briggs, C. J., Knapp, R. A., & Vredenburg, V. T. (2010). Enzootic and epizootic dynamics of the chytrid fungal pathogen of amphibians. *Proceedings of the National Academy of Sciences*, 107(21), 9695-9700.

DiRenzo, G. V., Langhammer, P. F., Zamudio, K. R., & Lips, K. R. (2014). Fungal infection intensity and zoospore output of *Atelopus zeteki*, a potential acute chytrid supershedder. *PLoS One*, 9(3), e93356.

Ellison, A. R., Tunstall, T., DiRenzo, G. V., Hughey, M. C., Rebollar, E. A., Belden, L. K., ... & Zamudio, K. R. (2014). More than skin deep: functional genomic basis for resistance to amphibian chytridiomycosis. *Genome Biology and Evolution*, 7(1), 286-298.

Ellison, A. R., Savage, A. E., DiRenzo, G. V., Langhammer, P., Lips, K. R., & Zamudio, K. R. (2014). Fighting a losing battle: vigorous immune response countered by pathogen suppression of host defenses in the chytridiomycosis-susceptible frog *Atelopus zeteki*. *G3: Genes, Genomes, Genetics*, 4(7), 1275-1289.

Voyles, J., Woodhams, D. C., Saenz, V., Byrne, A. Q., Perez, R., Rios-Sotelo, G., ... & Reinert, L. (2018). Shifts in disease dynamics in a tropical amphibian assemblage are not due to pathogen attenuation. *Science*, 359(6383), 1517-1519.

Weldon, C., Channing, A., Misinzo, G., & Cunningham, A. A. (2019). Disease driven extinction in the wild of the Kihansi spray toad (*Nectophrynoides asperginis*). *bioRxiv*, 677971.

We thank the reviewers for their comments. In this rebuttal letter we addressed all issues raised. We performed an additional in vivo trial to provide a proof of concept for our key finding.

Reviewers' comments:

Reviewer #1 (Remarks to the Author):

Greener et al. explore the effects and interactions of low and virulent isolates of *Batrachochytrium dendrobatidis* (Bd) and *B. salamandrivorans* (Bsal) on amphibians. The study addresses an important question, but unfortunately, this a classic case where the authors have chosen breadth at the expense of depth and rigor of experimental design. The manuscript includes field sampling (n=5), phenotypic characterization of multiple Bd isolates, infection trials of amphibians (n=3-6), tests of the protective capacity of low virulent Bd against high virulent Bd (n=6), Bd growth in amphibian mucosome (n=3), gene expression trials of Bd, in vitro infection of A6 cells (I have no idea what the sample size was because I don't believe it was provided), whole genome sequence analysis (unknown sample sizes because never provided), and trials on the protective capacity of Bd to Bsal in salamanders (n=6-9). Most of the sample sizes are either not provided or between 3 and 6 for the most salient studies on interactions between the fungi and the amphibians.

> We will respond to these potentially important, general remarks first before responding more in detail point by point. We disagree with the overall suggestion that this study has chosen breadth at the expense of depth. The current manuscript contains results of over 5 years of intensive studies. The general remark on the small sample size is for most cases detailed incorrectly. We do realize that one must take care extrapolating results based on 5-6 individuals. However, the number of animals used must not (according to European law) exceed the number of animals necessary to prove a given point. It is widely accepted that animals, if kept individually under identical laboratory conditions simultaneously, should each be considered replicates. If results obtained using the (indeed) relatively small number of animals yield significant results (as estimated before the onset of the trials, please see further under EU legal requirements for laboratory animals), these should not be discarded as "not meaningful" without any further arguments. As small group sizes do decrease the statistical power of the analysis, we must therefore aim at demonstrating relatively large effects at the individual level and we have done so, finding evidence that convincingly supports the following points:

- * Amphibian populations in our study area persist in the presence of BdGPL*
- * Midwife toads can be considered a European sentinel species for Bd-induced chytridiomycosis*
- * Midwife toads tolerate infection with local BdGPL isolates but are highly susceptible to the hypervirulent BdGPL isolate JEL423*
- * BdGPL pathogenicity correlates with Crinkler (CRN) gene expression pattern and host cell invasion*
- * Genomic variance associated with BdGPL virulence and host cell invasion*
- * Midwife toads that had been previously infected with the BdBE1 and BdBE3 local isolate showed significantly higher survival than those infected directly with BdJEL423*
- * Midwife toads, which have been pre-exposed to low virulent BdBE3 are protected against infection with highly virulent BdJEL423*
- * BdGPL reduces pathogenicity of Bsal in marbled newts, but not fire salamanders*

In other analyses, non-significant results may indeed reflect small sample sizes. However, we never interpret failure to reject the null hypothesis as evidence of no effect and make no conclusions for such cases. We provide all sample sizes and degrees of freedom, standard errors and/or confidence intervals for all estimates, and clearly provide all data both in the figures and appendices.

Please do notice the following concerns from our side:

- 1) We are strictly bound by EU regulations on the use of laboratory animals. We simply cannot use more animals if we are able to demonstrate a point using a smaller number. We sincerely hope Nature Communications will take ethical considerations into account.*
- 2) We did use, for all species, a unique collection of precisely staged specimen (e.g. within 10 days of metamorphosis). Assembling such collections is very challenging in itself. Please note that infection trials in e.g. midwife toads are notoriously more challenging than similar experiments in more "convenient" laboratory animals (e.g. Xenopus). This unfortunately means that experiments cannot be repeated easily (besides the questionable ethics and extreme difficulty of acquiring permission if a given point has already been reasonably proven).*
- 3) We did use species that only exceptionally can be used in experiments, refraining from using wild animals (which is not routinely permitted in the EU).*

In conclusion: we strongly disagree with the remarks of this reviewer regarding the majority of our trials where the reviewer is concerned with the small number of animals used. Apart from the lack of scientific need to increase the number of animals, ethical constraints should be taken into account.

Where we do agree with this referee and a point that may need improvement, is the trial in midwife toads, where we assess the protective capacity of the low virulent Bd. We do agree that the evidence is currently relatively weak. Since we agree that this part is key to the study, we repeated this experiment using a larger number of animals. For the original trial, we took advantage of our infected animals, that were re-exposed to the virulent strain. We agree that this is a rather opportunistic, yet meaningful and correct, approach. We therefore designed a novel experiment, using a larger number (40) of staged midwife toads (please see previous remarks: it is highly challenging to get a sufficient number of such species for experimental purposes, raised in an SPF environment). For statistical power, we again preferred to keep animals individually, under identical conditions. We reduced the number of treatments for this experiment to either pre-exposure to the low virulent BdBE3 or not. All animals were subsequently challenged with the highly virulent BdJEL423 and infection loads followed up. Results were very clear-cut: pre-exposure resulted to detectable Bd DNA at only two occasions, at low levels, in 2/20 animals. In contrast, all non pre-exposed animals built marked infections with BdJEL423. These results are highly significant and prove the point that previous exposure to low virulent BdGPL protects against infection with a highly virulent isolate. This experiment has been the reason of the long delay before responding to the reviewers.

Most of the statistical analyses are not significant because these sample sizes are highly insufficient for quantifying Bd or Bsal loads because these response variables have negative binomial error distributions. The rule of thumb for negative binomial error distributions is that 80% of the pathogen is found on 20% of the individuals, requiring much larger sample sizes than the authors chose to properly characterize responses.

> This comment might be based on a partial misunderstanding of our methods. While we agree that sample sizes are generally small (but see above our explanations for such limitations), only part of our

analyses considered infection loads as response variables. Following the reviewer's suggestion we have removed these from our methods and results. In all other cases, infection loads were either not considered, or used as predictors of binary outcomes (infection and mortality). For these analyses, although it is true that infection loads are generally overdispersed, their error distribution is not relevant to our choice of model (the GLM makes no assumption about the distribution of predictors) or conclusions in the way that the reviewer implies.

Moreover, the authors ignore the amphibians that did not get infected, probably to reduce variability and increase statistical power, but these resistant amphibians should not be ignored. This could greatly alter the conclusions. Sample sizes between 3 and 6 are even more insufficient for detecting effects on mortality, which is probably why most effects on mortality were not significant (possibly false negatives?).

> Please see further in the response letter. We think this is also based on a misunderstanding. We did include animals that did not get infected in the binomial infection/mortality analyses, since we did consider the possibility of e.g. resistance to be relevant. Following a standard two-part (hurdle) modelling approach, we then removed non-infected animals from analysis of load dynamics, precisely because their zero loads likely reflect different processes (e.g. resistance vs tolerance), thus inflating the number of zeros in the data.

Finally, the studies on the effects of low virulent Bd on higher virulent fungi confounded selection and phenotypic plasticity because the authors let considerable mortality occur from the first exposure to the fungi.

> We disagree with this remark. A single animal of all 29 toadlets infected with the low virulent strains died due to infection. This single animal will have had at best a marginal, if any contribution to selection for resistance during the experiment.

I also found it surprising that the manuscript ignored studies and theory on coinfections, given that this seemed to be a major focus of the current work.

> Studies that are pertinent to this paper were added.

In general, I found that the studies lacked necessary rigor to support the conclusions drawn by the authors, that the authors speculated too much in the Discussion, and that the authors did not acknowledge the many assumptions and caveats necessary to support their conclusions (e.g., no or weak host-by-parasite genotype interactions, no drift due to low sample sizes).

I do think there is a lot of valuable information in this manuscript, but for all the reasons just described, I cannot support publishing this work unless the authors can increase their sample sizes, reduce speculation, and better acknowledge the many important assumptions associated with their conclusions.

L 60: Your study doesn't quantify the costs of endemism. This was just a survey of Bd on amphibians. Please revise accordingly. Speculation.

> We toned down the conclusion of this paragraph and changed the title into: Populations persist in the presence of BdGPL

L 92-100: Sample sizes are way too small. Concluding that midwives toads are sentinel species is ridiculous. You have tested only two isolates of Bd and never provide a definition of a sentinel species. This also assumes no host genotype by parasite genotype interactions or at least the interaction is weak relative to the main effects. All of this is being concluded with a sample of five individuals per species, which is way too low given that Bd is distributed in a negative binomial manner (i.e., where most of the parasites are found on a small number of hosts)! Several studies suggest that samples sizes of 15 or higher needed to properly capture the distributional patterns of many negative binomially distributed pathogens.

> Five animals per species is indeed a low number but please see responses above for sample size limitations and load/mortality analyses. It has been a challenge to simultaneously obtain similarly staged animals of these, difficult to obtain species. While we do agree that small sample sizes mean more subtle and small differences are unlikely to be detected, these are not arguments to refute the significant findings that were obtained. We added a definition of sentinel species to the text. ‘... and reflects the European field situation^{1, 7, 15, 16, 27-29}. Based on these results we classify the midwife toad as a Bd sentinel (indicator of the presence of disease) species in Europe. We do protest the offensive language used by this referee. Sentinel is a common term in animal disease epidemiology and this being named ridiculous is neither justified nor productive.

L 105-109: But, if these data were collected from the same individuals, the trials are not independent are thus are simply autocorrelated. Was that the case?

> These data were collected from other individuals. We agree that this information was not clear from the materials and methods section and have corrected it in that section.

L 121-138: Based on what sample sizes? I could not find this in the Methods.

For the QPCR analysis we analysed the expression of certain genes in spores originating from JEL423; BD1; BD2; BD3; BD4; BD5; BD6; BD7; BD8; BD9; BD10 with and without tissue of Alytes obstetricans (A.o.). Every condition was analysed in fourfold and every sample was tested in threefold using QPCR.

In detail: RNA was extracted, cDNA was synthesized and gene expression was analysed from the following conditions:

	Condition	RNA ng/μl
spores	JEL423 A	186
	JEL423 B	254,6
	JEL423 C	156,8
	JEL423 D	210,2
spores + tissue	JEL423 E	95,2
	JEL423 F	153,9
	JEL423 G	131,6
	JEL423 H	133,7
spores	BD BE1 A	197,2
	BD BE1 B	190,2
	BD BE1 C	221,3
	BD BE1 D	174,9
spores + tissue	BD BE1 E	250,5
	BD BE1 F	199

	BD BE1 G	255,8
	BD BE1 H	191,8
spores	BD BE2 A	412,8
	BD BE2 B	372,7
	BD BE2 C	345,2
	BD BE2 D	208
spores + tissue	BD BE2 E	148,8
	BD BE2 F	298,3
	BD BE2 G	156,5
	BD BE2 H	250,1
spores	BD BE3 A	208,5
	BD BE3 B	163,2
	BD BE3 C	184,1
	BD BE3 D	155,6
spores + tissue	BD BE3 E	765,9
	BD BE3 F	1109,6
	BD BE3 G	1088,6
	BD BE3 H	1641,6
spores	BD BE4 A	194,5
	BD BE4 B	135,8
	BD BE4 C	108,3
	BD BE4 D	146,1
spores + tissue	BD BE4 E	345,2
	BD BE4 F	390,7
	BD BE4 G	197,5
	BD BE4 H	171,5
spores	BD BE5 A	110,2
	BD BE5 B	1112,7
	BD BE5 C	76,3
	BD BE5 D	77,6
spores + tissue	BD BE5 E	319,6
	BD BE5 F	273,3
	BD BE5 G	568,4
	BD BE5 H	574,4
spores	BD BE6 A	282,9
	BD BE6 B	218,9
	BD BE6 C	519,4
	BD BE6 D	244
spores + tissue	BD BE6 E	96,4
	BD BE6 F	333,6
	BD BE6 G	105,2
	BD BE6 H	234,5
spores	BD BE7 A	260,8
	BD BE7 B	372,5
	BD BE7 C	173,7

	BD BE7 D	157,8
spores + tissue	BD BE7 E	268,5
	BD BE7 F	343,6
	BD BE7 G	353,3
	BD BE7 H	288,6
spores	BD BE8 A	179,8
	BD BE8 B	112,4
	BD BE8 C	97,3
	BD BE8 D	142,9
spores + tissue	BD BE8 E	312,7
	BD BE8 F	298,8
	BD BE8 G	346,4
	BD BE8 H	254,5
spores	BD BE9 A	85,2
	BD BE9 B	122,2
	BD BE9 C	118,5
	BD BE9 D	154,8
spores + tissue	BD BE9 E	137,9
	BD BE9 F	86,2
	BD BE9 G	107,2
	BD BE9 H	199,3
spores	BD BE10 A	120,2
	BD BE10 B	173,7
	BD BE10 C	223,1
	BD BE10 D	176,7
spores + tissue	BD BE10 E	463,4
	BD BE10 F	432
	BD BE10 G	535,7
	BD BE10 H	435,7

To make this more clear to the readers, we updated:

- the information in the materials and method section (L660-669) by clearly indicating the number of biological replicates ($n = 4$) and technical replicates ($n = 3$).
- the legend of figure 2 by mentioning the number of biological replicates ($n = 4$)
- For the A6 invasion model, we agree that the information about the sample size was lacking. Every condition was tested in threefold and three independent repetitions of the experiments were conducted.

To make this more clear to the readers, we updated the information in the materials and methods section (L 713) by mentioning “To assess the *in vitro* infection dynamics of different BdGPL isolates, three independent *in vitro* experiments were conducted with every condition being tested in triplicate.”

L 136-138: I don't understand these statistics? What do CRN_#s mean? CRN is never defined in the manuscript. Also, what are the sample sizes? They are never provided.

- *We compared gene expression of a selection of virulence genes as identified by Farrer et al. (2017). This is clearly stated in the materials and methods section and we refer to supplementary table 10 for an overview of the tested genes.*

Supplementary Table 10. List of genes and sequences of the primers used for quantitative PCR analysis.

Gene	Forward primer	Reverse primer	Ref
α-centractin	GCAGCATGGAGTTGCTACTG	AGCTTGGTCACGATTGGAAC	Farrer et al., 2017
APRT	GGTTGCCACTTGGAGTCTGT	ATGGCTGGATGGAAACTCTG	Verbrugghe et al., 2019
TUB	CTCTCGGTGGTGGTACTGGT	AGGGTATTCTCCGCGAATCT	Verbrugghe et al., 2019
Ctsyn1	TCCTCAGCAGCTCCTATTCG	CTCGACGTCTTTTCAGGA	Verbrugghe et al., 2019
ADM_20379	CTGGTATGGACGCTCTCGTT	AGACTAAGCCAGTCGCTCCA	Farrer et al., 2017
ADM_23888	CACCCAACGAGTTC AAGGTT	GCCGCTTGGATATGGACAG	Farrer et al., 2017
ADM_27285	GCAGATGGTCAACCTGGAGT	CCACTGTCTCGATTGGATT	Farrer et al., 2017
ADM_25206	GGTCTTGACAGCCAAATCGT	CTCTAGCCTCACCGTCGAAC	Farrer et al., 2017
ADM_22285	GGTGTCTCAGGTCGGTTGAC	GATTCTTGGCAGACACGAT	Farrer et al., 2017
CBM18_20255	ATCTTGCTTGACACCCGAAG	GTGACTTGGCTGATGCCTTT	Farrer et al., 2017
CBM18_28751	CGTGTGGACGTCGATACAAC	CATCCAACACTGCAATGAGC	Farrer et al., 2017
CBM18_26087	TCGTGAACTAACGCAACAGC	CAGACGGTACTTGACGCAGA	Farrer et al., 2017
BDEG_24342	CCGGCTACAAGCTTGTGAGA	GTGTTGGATCCAGGACCCTG	Farrer et al., 2017
BDEG_26151	CAGCTGATGAAGATGGCTCA	GGTTCGTTAGTCGGGACAGA	Farrer et al., 2017
CRN_23176	AAACGCCCTTCGCTTCGATA	TCTTTCTCCAAGCTGAGCGG	Farrer et al., 2017
CRN_25085	CTCCCGGTTGACATCACA	GAACAGCGAACCACAGCTTG	Farrer et al., 2017
CRN_22492	CTCCCGGTTGACATCACA	GAACAGCGAACCACAGCTTG	Farrer et al., 2017

Farrer, R. A. et al. Genomic innovations linked to infection strategies across emerging pathogenic chytrid fungi. *Nat Commun* **8**, 14742 (2017).

In the paper of Farrer et al. (2017) they describe a notable expansion of CRN-like genes in Bd and CRN_23176, CRN_25085 and CRN_22492 were shown to be of importance during the initial contact with the host.

Farrer et al. 2017 Fig. 4C: CRN mean fold changes in mRNA expression profile in *Bd*. The data show the normalized target gene amount in spores that were incubated with skin tissue of Tw for 2 h, 3-day-old sporangia grown in TGhL and skin tissue from chytrid-infected Tw animals relative to freshly collected spores which is considered 1.

As such, we use the nomenclature of the genes as mentioned in the original paper of Farrer et al. (2017) and “CRN_#s” reflect these genes. We do agree that we should include the references of Farrer et al. (2017) after L231-232. The abbreviation Crinkler and Necrosis (CRN) gene is mentioned in L215-216.

Correlation between the relative expression of each isolate to *BdJEL423* for CRN-like genes of fresh spores exposed to midwife toad skin and colonisation capacity from the individuals from the multi-isolate *A. obstetricans* infection trial was assessed by Spearman’s rank correlation in R. This was clarified in the manuscript.

L 171-174: Nothing is significant because a sample size of six toads is way too low.

> Significance was clear in several instances, particularly for mortality effects, and we never infer lack of significance to imply lack of effect. However, we do agree with the reviewer that in general the number of animals per *Bd* isolate is low, and this is a potentially weak point. This experiment is repeated using a larger group and a single low virulent *Bd* isolate. We summarized the results of this trial above.

L 177-187: These studies confound selection with plasticity because the infections were allowed to induce significant mortality.

> Mortality was only observed in animals exposed to *Bsal*, as expected.

L 183-185: Again, nothing is significant because the sample sizes are insufficient to detect effects on mortality.

> As explained above, we have removed the results for the analyses where infection loads were taken as response variables. We have opted to keep the other results, including those for *Pleurodeles*, although indeed the reviewer is correct that these results are inconclusive, likely due to small sample size. We added this in the discussion (we would prefer not to leave these results out). Results in fire salamanders and marbled newts are clear and the conclusions are sound and in line with other published evidence. Thus we believe that results and conclusions will not benefit substantially from increasing numbers of experimental animals for the latter two species, but would for the ribbed newts.

New paragraph in discussion: Reduced mortality and delayed time to death demonstrate that prior exposure to Bd reduces pathogenicity of Bsal in marbled newts. This may equally pertain to the ribbed newts, in which Bsal mortality occurred only in newts that were not previously exposed to Bd. However, low mortality rate and relatively small sample size hamper meaningful statistics for this species. In contrast, all fire salamanders succumbed to Bsal infection within a similar timeframe, regardless of prior Bd infection. The host species dependent protective effect of a pre-existing Bd infection against Bsal may present as a double-edged sword for the amphibian community. The reduced pathogenicity in the marbled newts coincides with a longer infectious period and high Bsal infection loads, which is likely to facilitate Bsal transmission to susceptible animals. Increased transmission opportunities may offer an additional explanation for the local extirpations observed in Bsal-infected fire salamander populations in parts of Europe where Bd is widespread.

L 194-196: Not sufficient sample sizes to conclude this.

> We strongly disagree; results obtained in the lab trials not only are corroborated by our own field observations, but are in line with general findings across Europe. We do not see any valid reason to doubt these conclusions.

L 200-203: So perhaps this is not evidence of Bd genotype changes but evidence of host species variation?

> We do not understand this comment. The midwife genotype used is the northern one, from populations that are currently not in decline. All individuals came from a single lineage and were randomly assigned to groups. These midwife toads (as metamorphs) were demonstrated to be highly susceptible to the highly virulent isolate. Of course we expect some variation in individual host response (and indeed see that within treatment groups) but statistically the difference is greater between treatment groups than the individual variation shown within groups, and so it is reasonable to assume this difference is due to the treatment.

L 204-209: I don't agree with any of this because the authors have completely ignored the possibility of g x g interactions. Perhaps state that, assuming no parasite genotype-by-host genotype interactions...

> For some pathogens the interaction of host and pathogen genotypes have been shown to be significant for host susceptibility (e.g. Lambrechts et al., 2005). We do not have a single indication of genotype dependent susceptibility to Bd in Europe for the moment, let alone for genotype genotype interaction. Whilst such interactions may also exist for BdGPL, as all experimental animals came from single isolated colony, thus specimens could reasonably expect to be genetically similar and were randomly assigned to treatment groups we aimed to minimise the influence of such interactions on our interpretation of Bd isolate virulence.

We do have strong evidence for the role of the environment (see eg Schmeller et al., 2014) and this paper does provide strong arguments for Bd virulence as determinant for the scenarios we currently witness in Europe.

L 212-215: By stating "would depend", this is stated as if your study was causal. Your study was only correlational. You don't know that these differences were the cause of the different susceptibility.

> *That is correct and we adapted this paragraph in the text. 'However, pre-existing low virulent Bd isolates may at least initially affect disease dynamics of an invading hypervirulent Bd. Their protective effect is associated with several traits that correlate with their colonisation ability and invasiveness: expression and function of Bd crinkler genes, metalloproteases and other effector proteins.'*

L 218-220: Unjustified speculation

> *this is removed from the text.*

L 221-224: These studies confound selection with acquired resistance and thus cannot say anything about acquired resistance at all. They were not done in a manner where the infections were cleared before substantial mortality and thus we do not know whether any susceptibility differences were due to selection or plasticity. Another example of where the authors attempt to do too much in my opinion in a single study without doing many of the experiments well (insufficient sample sizes, rigor, experimental design, and controls).

> *With regards the reviewer's conclusion this may be selection due to substantial mortality, again, in those isolates hypothesised to confer some degree of resistance only 1 in 29 animals died and so we would again argue it is not an example of selection, and most likely reflects acquired resistance. The underpinning mechanism is indeed not clear and thus we refrain from any conclusion regarding any such mechanism.*

L 224-227: Again, way too much speculation. Conduct the experiments or develop the mathematical models rather than excessively speculate on how these dynamics will play out.

> *This is speculative indeed but we feel this contributes to the discussion and ensure it is clear from our wording that this is speculative.*

L 228-229: Not an explanation, a hypothesis! Please be more precise with your language. You haven't explained these extirpations at all. You simply offer a hypothesis.

> *We added "may offer" since indeed this is hypothetical. '*

L 232-234: I don't find this result convincing at all for two reasons. First, insufficient sample sizes and second, it is unclear whether the results are due to selection or plasticity.

> *Please see earlier responses: selection is not an issue here, given that only one animal died after exposure to the low virulent isolates.*

L 234-236: This study did not test the hypothesis that this increases risk to highly susceptible hosts. This should be stated as a hypothesis that deserves testing. Please make this clearer.

> *We consider the wording ("may increase") to be clear that this is hypothetical. Also, this is a well-known phenomenon in epidemiology: amplification of reservoir dynamics.*

L 236-238: Perhaps, but this assumes that either 1) there is no host genotype-by-parasite genotype interaction or 2) that this interaction accounts for much less variation than the main effect of

parasite genotype. We have no evidence to support either of these assumptions to which I am aware. Thus, I cannot advocate this overall conclusion.

> Please see our previous responses on the gxg interactions. This very paper provides a very important nuance to the currently widely accepted concept that BdGPL is highly virulent, which in itself hinders any active conservation throughout Europe (given its omnipresence in European systems). At least for the European situation, our conclusions make sense and we are willing to scale down the overall conclusion to the European continent. Optimizing diagnostics may indeed incentivize active mitigation (which is currently simply not done in the EU).

L 334: Five individuals are not enough to draw conclusions about the population given that loads are distributed as negative binomial error distributions, i.e., most of the zoospores are found on a very small number of individuals. And, five individuals will unquestionably be insufficient to detect effects on mortality.

> This experiment was performed to detect which European amphibian species is the best to use in isolate pathogenicity comparison trials. As mentioned in the materials and methods section, all individuals were housed individually and infected in a standardized way – this is not a wild population (where indeed 80% of the pathogen may be found on 20% of individuals). As explained above, we have taken onboard the reviewers' concerns about infection load distribution and thus removed the part of the analysis where infection loads were considered as the response variable here. However, other analyses such as the proportion of animals that died are still valid. Obviously, the small sample size will reduce the statistical power of the test; we recognize this limitation, provide the relevant information, and do not assume lack of significance to imply lack of effect. However, we also note the striking difference between the local low virulent (1/29 mortality) and high virulent (9/9 mortality) strains.

L 352-354: Why are you ignoring individuals that did not become infected? These individuals are important in assessing resistance of the population. This is a big issue for me.

> We think this probably reflects a misunderstanding. We did not ignore individuals that did not become infected. Species where all animals were still negative in qPCR three weeks after Bd exposure were considered not infected and removed from the experiment. Absence of infection was confirmed by histopathology and immunohistochemistry. As explained above, we did include animals that did not get infected in the binomial infection/mortality analyses, since we did consider the possibility of e.g. resistance to be relevant. As is common practice with zero-inflated, overdispersed data we conducted a standard hurdle modelling approach, where we then removed non-infected animals from analysis of load dynamics, precisely because their zero loads may reflect different processes (e.g. resistance vs tolerance).

L 361-362: Six animals is not enough per treatment group to assess time of death. Sample sizes are way too low.

> In general terms, we obviously agree that a sample size of six will have low statistical power. See above our explanation for the reduced sample size. However, in this case time of death was not even explicitly modelled, because of the obvious difference between the groups: the JEL423 group (nine animals) had 100% mortality within a month, and only one of the other 29 toadlets died. We simply

report the mean and range of the time to death for the JEL423 group, and the mean can of course be calculated from nine individuals – although uncertainty will be wide. In the other groups, with zero mortality, the mean was not calculated (see Table S5).

L 365: What is "a hide"?

> We meant a hiding place. This is adapted in the manuscript.

L 368: What were the humane endpoints that induced euthanasia? Your study needs to be replicable.

> the humane endpoint are mentioned in the materials and methods section – infection trials.

“Humane end points were set at the loss of self-righting ability and/or change in posture.” Animals were euthanized by an overdose of pentobarbital.

L 369-371: Huh? This is a really weird change in methods that makes the results difficult to interpret. Again, a sample size of 5 and 3 is simply insufficient for time of death trials.

> For each isolate, at least 6, and for isolate JEL423 and BdBe5 9 and 11 toadlets were used respectively.

L 381-383: The problem with this approach is that you have induced a selection event given that there was considerable initial death. Given that the sample sizes are extremely small, any initial death, whether it be due to treatment or stochastic, will increase the chances of drift as well. This is all too problematic for me to feel confident in any of the conclusions.

> A single animal out of 29 died due to low virulence isolate Bd infection. A second animal due to an accident. We do not see how this could possibly contribute significantly to selection. If this is indeed one of the major issues of this reviewer not to agree with the conclusions, we strongly disagree with this reviewer’s proposal on the decision regarding this manuscript. This is simply not correct.

L 394-395: A Weibull distribution does not account for censoring. Please revise to improve accuracy.

> we agree this was unclear. We have changed to “...a Weibull distribution while accounting for censoring”.

L 395-397: You did this because you don't have adequate statistical power because your sample sizes are too low across all experiments.

> Here we agree with this reviewer and as mentioned, performed an additional experiment. Please see above.

L 398-400: You are trying to do way too much in this study. As a consequence, you have done nothing well. Sample sizes are simply too low.

> Before making such statements, please provide better arguments than the generalisations used throughout this review. We strongly disagree here and would ask the editor not to take such offensive and unsubstantiated comments into consideration.

L 400-401: A sample size of three for the mucosome experiment is completely insufficient.

This experiment was repeated with ten animals per species. Results are provided in the manuscript.

L 447-478: What are the sample sizes for the in vitro infection of A6 cells? I can't find it documented anywhere.

We agree that the information about the sample size was lacking in the manuscript, we have corrected this and thank the reviewer for bringing this to our attention. Every condition was tested in threefold and three independent repetitions of the experiments were conducted.

To clarify this for the readers, we updated the information in the materials and methods section (L 713) by mentioning: "To assess the in vitro infection dynamics of different BdGPL isolates, three independent in vitro experiments were conducted with every condition being tested in triplicate."

L 508-511: Sample sizes are 6-9, which is better than the other studies, but still insufficient for detecting effects on mortality, especially for cases with 6 animals.

> we disagree with the reviewer. While we acknowledge that small sample sizes limit the statistical power and mean only large effects may be detected, here meaningful, relevant and statistically significant results were obtained.

L 511-515: Why these treatments in these orders?

> We acknowledge this may be unclear. To clarify we have rephrased the paragraph: 'Two groups of each species were exposed to local BdGPL (1ml of BdSP11 10^6 spores for 24 hours); three weeks later one of these groups was exposed to Bsal (1ml of AMFP18/02 10^3 spores for 24 hours). At this 21 day time point, a Bsal control group of each species was also exposed to the Bsal inoculant (1ml of AMFP18/02 10^3 spores for 24 hours).'

Figure 1: How is fecundity different than number of zoospores? Please define here given that figures and their legends are supposed to be stand alone.

> the following definitions are added to the figure legend: N of sporangia = number of sporangia in the central 1000 x 1000 pixels; area of sporangia = area of the largest 10 sporangia; N of zoospores = number of spores in the central 1000 x 1000 pixels of the well/image; fecundity was calculated following the formula (Average $N_{ZOOSPORE}$ / Average N_{SPOR}) / Average A_{SPOR} . BdBE1-10 are Belgian isolates isolated in this study. JEL423 is BdGPL isolate isolated from an episode of amphibian mortality in the neotropics¹⁶.

Figure 3: Cool, but there are just too few toads. The sample sizes are too low. You are comparing 5

deaths to 1 or 2 deaths. Chance alone can explain these results. This is why nothing was significant in the Results section.

> Please see previous answers.

Reviewer #2 (Remarks to the Author):

What are the major claims of the paper?

In this manuscript, Greener et al. describe a study that investigated infectious disease dynamics of fungal pathogens of amphibians. The investigators integrated field surveys and multiple experimental investigations in an attempt to understand the importance of pathogen endemism in determining disease dynamics in a specific amphibian host assemblage. Specifically, they aim to address the question of why or how the infectious disease chytridiomycosis has had different impacts on amphibians around the world.

The authors' primary claims are that 1) the fungal pathogen Bd is co-existing with amphibian hosts in Europe (Belgium), with no strong evidence of recent disease-induced declines, and 2) the presence of low-virulence Bd (in this case, Bd isolates that originated in Europe, which they term "endemic") may protect, or provide a barrier, for amphibians against more virulent Bd (which they term "exotic") invasion.

While these central claims are intriguing, I find that they are an odd fit for an otherwise highly compelling study. As I detail in my review below, I believe many aspects of this study are remarkable in terms of their integration, rigor, and scope to understand the pathophysiology of chytridiomycosis and the differential pathogenicity of Bd isolates. On this count, the authors provide numerous insightful and novel findings that will advance our understanding of this disease and be of interest to a broader audience. However, I am unsure if the strong points of this study provide robust support for their central claims as outlined in the paper. Below I provide some additional detail on why their claims regarding Bd endemism may benefit from additional work and/or clarification.

Is the work convincing?

To reiterate, I find this study to be convincing and exciting for multiple reasons, but they are reasons that differ from those that the authors have articulated. To begin with, I believe they have captured the molecular and cellular pathophysiology of chytridiomycosis in a way that is novel and highly effective. While previous studies have used similar experimental designs in a rather piecemeal fashion, this study integrates multiple innovative approaches in an elegant way and has applied them to multiple host species. The insights provided on this topic will be of great interest to a diverse community of researchers. In addition, I believe this study provides a detailed mechanistic explanation for differential virulence among Bd isolates. The experiments were very well executed and I find them to be highly convincing. These aspects of the study address long-standing questions that are critical to resolving the devastating impacts of this disease. These accomplishments are to be lauded and I expect they will significantly advance this field of study.

With that said, I am less enthusiastic about some of their central assertions regarding 1) why host assemblages differ in their responses to Bd emergence and 2) that infection with a low-virulence isolate provides a barrier to invasion with a more virulent isolate. In fact, I would strongly discourage the inclusion of several of their inferences for reasons that I outline below.

If not, what further evidence would be required to strengthen the conclusions?

1) The authors observed different disease outcomes following exposure to multiple Bd isolates (e.g., Fig 3). They attribute these findings to low-pathogenicity due to isolate endemism (endemic European isolates) and suggest that this may be why impacts of the disease vary in Europe (lines 48-

50 and 189-191). Their rationale is primarily based on their finding that the rates of mortality were lower in European amphibian hosts when exposed to European Bd isolates compared an isolate from Central America. While I agree that they have successfully documented differential pathogenicity, it is unclear to me that this is due to the geographic point of origin of the isolates (i.e., isolate endemism) and/or that the results can explain the variable disease impact in Europe. Here are two additional pieces of information that should be considered and additional suggestions that may bolster this particular argument:

- Pathogenicity can shift in vitro. We know that pathogen virulence (including for Bd) is malleable, dynamic, and can change depending on a wide range of factors, including isolation and passage methods, media, incubation temperatures, etc. As such, it would be helpful to know if all of the isolates used in this study have always had the same (or comparable) in vitro maintenance. Furthermore, there are methods to mitigate the effect of differences in pathogen maintenance. For example, using an approach where all of the isolates are cryopreserved, revived, and subsequently passaged identically in the laboratory could address this issue. If the authors considered this question, and used methods to address this prospective problem, they are not currently presented in the paper. Without this information, it is reasonable to consider that the results were due to treatment of the isolates prior to the experimental work, rather than the geographic point of isolation.

> we agree with the reviewer. To overcome this problem all isolates used in this study had the same in vitro maintenance. All isolates were preserved in liquid nitrogen at passage 10. Before use in an experiment, isolates were passaged one time in TGhL broth at 20°C. This information is added to the materials and method section.

- Differential pathogenicity, not endemism, may be the most parsimonious explanation. We have known for many years that different Bd isolates, even from within the same strain/lineage, can have variable disease outcomes. As such, it is possible that the authors have documented different levels of pathogenicity that are unrelated to the geographic origins of the isolates. In other words, it may not be appropriate to explain their results through the lens of exotic versus endemic isolate pathogenicity. One way to address this question would be to include parallel group of "exotic" host amphibians in their exposure experiments (e.g., using a non-European, neotropical host species). This approach would provide a fully factorial study design that more directly examines "exotic versus endemic" patterns. If an exotic (Central American) species had been similarly exposed to all isolates, the results might have revealed an opposite pattern, where the exotic hosts exhibited high mortality when exposed to the European isolates (* but see note below). Such a result would support their central claims. In addition, they could have included additional "exotic" Bd isolates to provide biological replication. Both of these modifications to the experimental design might have provided more direct evidence to support their argument that the European isolates were less pathogenic due their endemism (and therefore help explain disease dynamics in Europe).

> We agree with this reviewer to tune down conclusions to pathogenicity and not endemism per se and adapted this throughout the text. The main point we wanted to demonstrate is that the Bd isolates that are currently present, are of (relatively) low virulence, which is in line with observed disease dynamics in the region.

*As an additional note, my suspicion is that this particular "exotic" isolate is also highly pathogenic to amphibian species in Central America. There is evidence for this in the literature (DiRenzo et al. 2014,

Ellison et al. 2014a and 2014b, Voyles et al. 2018) where this isolate has caused up to 100% mortality in Central American and North American amphibian host species.

Without this additional information, it is not clear that we can rule out the possibility that JEL423 is simply a more pathogenic isolate or has become one over the course of laboratory maintenance. While we may be able to conclude that the European isolates are less pathogenic (at this point in time) by comparison, it is not clear to me that this finding is wholly due to the fact that they come from Europe. They may simply be inherently less pathogenic.

> *We agree and this remark is in line with previous remarks regarding endemism and pathogenicity.*

Taking all this into account, I suggest that this conclusion statement may require some additional consideration and adjustment:

"Although Bd has caused death and species extinction on a global scale, its impact in Europe is limited. This can be explained by the fact that a limited number of European amphibian species are susceptible to Bd-induced chytridiomycosis, and by the presence of low virulence endemic BdGPL lineages that can protect susceptible species against virulent exotic BdGPL." (Lines 189-192)

> *We adapted to:*

"Although BdGPL has caused death and species extinction on a global scale, its impact in Europe is currently limited to specific regions. Our results suggest that low virulence BdGPL isolates co-occur with amphibian communities in the absence of population declines and that only a limited number of European amphibian species may be susceptible to the highly virulent isolates associated with massive declines elsewhere."

Not only is the importance of isolate endemism questionable (for the reasons explained above), but this explanation also fails to acknowledge the many spatio-temporal factors that strongly influence disease impacts (e.g., time since pathogen emergence, local and regional ecological factors, community composition, etc.).

> *We have given this reflection of this reviewer serious consideration within the author consortium and agree. Actually, we did not mean to bring the message that because of their endemism, the isolates used are of low virulence. Rather, our main conclusion is that BdGPL at a very local scale is quite diverse (see genotyping, isolates belonging to different clades within BdGPL) and of low virulence, compared to a known highly virulent isolate that is associated with the dramatic declines in Latin America. Virulence of local Bd isolates may provide a plausible explanation for the obvious differences in current disease patterns observed across Europe. We therefore adapted our text throughout and removed reference to any causative link between endemism and low virulence. We recognise that while Bd presence in the study area could be defined as endemism, the reviewer is correct that we cannot infer such statements at isolate level.*

Additional comments-

It is possible that addressing the points outlined above could alleviate several issues. However, I wanted to point out some additional concerns with the following terms: "endemic" and "exotic" Bd and "lineage" or "strain" vs. "isolate"

> we adapted the use of endemic, exotic, lineage, strain throughout the manuscript following the reviewer's comments

I suggest that the approach that the investigators used is essentially a comparison of multiple isolates from within the same BdGPL clade (lines 82-83). As such, it is probably inappropriate to use terms such as "BdGPL lineages" (lines 191-192) as it misrepresents the fact that they are all BdGPL isolates. This comment raises a slightly larger issue regarding the merits of comparing pathogenicity among the European isolates (e.g., BdBE1, BdBE3, BdBE4, BdBE5; Fig. 3). Because these isolates appear to be the same in terms of genetics (and most phenotypic characteristics), one could argue that the exposure experiment reflected a form of pseudoreplication (i.e., multiple experimental groups with the same genetic strain). However, I am not making that argument because 1) I noted that these isolates come from different amphibian host species and/or life stages (Supplementary Table 4), and 2) the strength of their pathophysiology findings are a result of the documented variation among isolates (possibly reflecting a subset of their phenotypic characteristics, such as number of sporangia; Fig. 1). Because these are not different Bd lineages, but rather multiple isolates of the same strain, perhaps a more accurate explanation is that the isolates used in this experiment were the most different in terms of phenotype and/or host origin/life stage. This terminology would be preferable to describing the isolates as different lineages. Line 389 and line 316-317

> We would like to stress that there is surprising genetic and phenotypic variation among the local isolates, not in line with a recent, single introduction event with subsequent clonal expansion (please see the O'Hanlon paper that depicts topology of the isolates used here in the Bd phylogeny, the isolates used belong to several clades and are not each other's closest relatives). Therefore, we do not consider this pseudoreplication. We do agree that lineage should be replaced by isolate. We added this information in the Materials and Methods section. "Line 316 six isolates were genotyped and their topology depicted in the global Bd phylogeny in O'Hanlon et al. (2018). These isolates belong to three clades."

2) The investigators suggest that infection with a low-virulence isolate provides a barrier to invasion with a more virulent isolate. Their support for this claim comes from the sequential exposure of toads to the low pathogenicity isolates (the European isolates) and subsequently to the high pathogenicity isolate (the Central American isolate) as well as a similarly designed experiment with salamanders. I have three concerns regarding part of the experiment:

- Regarding the methods and experimental design, it is unclear to me if the investigators controlled for the infection load at the time of the exposure to JEL423? I imagine that, with this study design, the investigators may have created a scenario for super infection (i.e., within host competition between two isolates). As such, the infection intensities may reflect the success of one or the other of the isolates. In order to evaluate this possibility, it would have been helpful to include parallel groups that received secondary exposures to the European isolates only (as a positive control) or if there was a way to distinguish between the isolates using a discriminatory PCR approach.

> We agree with this comment. The reviewer is correct that we cannot attribute observed mortality or infection dynamics to any of the two isolates used (there is no discriminatory PCR available, unfortunately). One of the major bottlenecks of these experiments, was the availability of a sufficient

number of staged midwife toadlets and we had to choose between including more treatments (such as a secondary exposure to the low virulence isolate) or more animals per treatment. The very nature of our experiments (making use of difficult to stage experimental animals, available in small numbers) prompted us to maximize the number of animals per treatment (still low). Indeed, we have to limit conclusions to overall Bd infection load. Since these were systematically lower than those in the highly virulent JEL423 single infection group and disease was tempered, we can conclude that exposure of midwife toads to the highly virulent isolate resulted in tempered Bd disease dynamics if pre-exposed to the low virulence isolate. We abstain from attributing the loads to any of both isolates used.

- Upon evaluation of the data provided in Supplementary Table 8, it appears to me that only one group of toads exposed to a low-virulence isolate had a significantly higher survival rate (BdBE3). This result concerns not only because the sample sizes are small, but also because it appears that only 1/4 of the groups with prior exposure to a low-virulence isolate had increased survival. I also considered the results from the exposure of salamanders to Bd and subsequently to Bsal, which were inconsistent across experimental groups. To me, the take-away message from these experiments is that the results of prior infection with a low-virulence isolate were mixed.

> We again agree and this is mentioned also in the discussion: the outcome depends on the isolate used and we correlate the extent of protection conferred to colonization ability of the low virulence strain used. The abstract was rewritten : "Pre-exposure to some of these low virulent isolates confers protection against infection with highly virulent BdGPL in midwife toads (Alytes obstetricans) and alters infection dynamics of its sister species B. salamandrivorans in newts (Triturus marmoratus), but not in salamanders (Salamandra salamandra)."

- The claim that a low-virulence isolate provides a "barrier" to subsequent does not appear to be fully supported. The subsequent exposures resulted in infection, and the maintenance of infection over time, rather than a resistance to infection. So, while it is not clear to me how infection operated as a barrier, perhaps the authors have a different interpretation of this term? If so, the use of this term may require some additional explanation.

> We agree with the reviewer. This sentence was deleted from the text based on previous comments and to avoid misinterpretation.

In raising these concerns, I would like to note that the accurate interpretation/presentation of these results will be highly consequential for conservation and management practices. To be sure, understanding the dynamics underpinning co-infection is incredibly interesting and exciting for academic reasons. However, I urge caution, especially with suggestions of "natural vaccination" (lines 233-234), for several reasons: we still have a limited understanding of co-infection processes, pathogen virulence can shift, and the repercussions of misperceptions in management scenarios could be grave. For example, the authors suggest that infected populations "might be protected against disease" (in lines 210-212). This statement is both speculative and alarming because, taken out of context, it could be used for a wide range of conservation actions (or inaction) that could be detrimental to amphibians based on our understanding of chytridiomycosis globally.

> We omitted any reference to the application of Bd as natural vaccine (this was not our intention).

While we agree that phrasing is crucial, we would like to keep some reference to the potentially protective effect of low virulent Bd in the manuscript. We propose to replace the sentence (line 240) by: "...may be less affected."

Do you feel that the paper will influence thinking in the field?

Potentially. While I am very enthusiastic about the core scientific content of their work, I am less certain of the "packaging" (i.e., their explanations and central claims). I believe they have not showcased their results in a way that fully captures how unique and exciting their study is. Specifically, they uncover critical mechanistic processes of a notorious disease system. Their findings are revelatory for the community of researchers working on chytridiomycosis as well as a broader audience interest in emerging infectious disease.

Minor comments-

Throughout the paper- The term "pathogenicity" or "pathogen virulence" may be preferable because in many fields, the term "virulence" is a composite trait, involving the host and the pathogen. Please see comments above regarding terms such as "lineage".

> *"Virulence" was replaced by "pathogen virulence" throughout the text.*

Lines 25-28 The opening sentence is rather long and the subject of the sentence is narrowly focused. Might it be better to use a sentence that appeals to the broader scientific community?

> *To appeal the broader scientific community, this sentence is replaced by: "Wildlife diseases are contributing to the current Earth's sixth mass extinction. One of these, chytridiomycosis, has caused mass amphibian die-offs. While global spread of a hypervirulent lineage of the fungus *Batrachochytrium dendrobatidis* (BdGPL) causes unprecedented loss of vertebrate diversity by decimating amphibian populations, its impact on amphibian communities is highly variable."*

Line 28 Is "populations" the best term? I suggest "communities" or "assemblages".

> *Based on the previous comment we changed the sentence to: "Wildlife diseases are contributing to the current Earth's sixth mass extinction. One of these, chytridiomycosis, has caused mass amphibian die-offs. While global spread of a hypervirulent lineage of the fungus *Batrachochytrium dendrobatidis* (BdGPL) causes unprecedented loss of vertebrate diversity by decimating amphibian populations, its impact on amphibian communities is highly variable."*

Line 30 and 61-62 The term "widespread" is pretty subjective and seems odd given that the results seem to be spatially limited to Belgium.

> *We agree with the reviewer. "Widespread" was deleted from both sentences. In the abstract Line 30 was changed into: "We combine field data with in vitro and in vivo pathogenicity trials that demonstrate the presence of a markedly diverse variety of low virulent isolates of BdGPL in northern European amphibian communities." The sentence on line 61 was changed into: "Opportunistic sampling of 1,483 amphibians for Bd in 2015-2016 revealed the presence of BdGPL in 17 out of 63 locations and 5 out of 7 species sampled in Flanders, Belgium (Supplementary Fig 1)."*

Lines 45-46 I believe this sentence does not reflect some well documented Bd-induced declines in North America (e.g., Briggs et al. 2010), Europe (e.g., Bosch et al. 2001), and Africa (e.g., Weldon et al. 2019)

> *We added the references to Briggs et al., 2010 and Weldon et al., 2019 and Bosch et al., 2001.*

References

- Bosch, J., Martínez-Solano, I., & García-París, M. (2001). Evidence of a chytrid fungus infection involved in the decline of the common midwife toad (*Alytes obstetricans*) in protected areas of central Spain. *Biological conservation*, 97(3), 331-337.
- Briggs, C. J., Knapp, R. A., & Vredenburg, V. T. (2010). Enzootic and epizootic dynamics of the chytrid fungal pathogen of amphibians. *Proceedings of the National Academy of Sciences*, 107(21), 9695-9700.
- DiRenzo, G. V., Langhammer, P. F., Zamudio, K. R., & Lips, K. R. (2014). Fungal infection intensity and zoospore output of *Atelopus zeteki*, a potential acute chytrid supershedder. *PLoS One*, 9(3), e93356.
- Ellison, A. R., Tunstall, T., DiRenzo, G. V., Hughey, M. C., Rebollar, E. A., Belden, L. K., ... & Zamudio, K. R. (2014). More than skin deep: functional genomic basis for resistance to amphibian chytridiomycosis. *Genome Biology and Evolution*, 7(1), 286-298.
- Ellison, A. R., Savage, A. E., DiRenzo, G. V., Langhammer, P., Lips, K. R., & Zamudio, K. R. (2014). Fighting a losing battle: vigorous immune response countered by pathogen suppression of host defenses in the chytridiomycosis-susceptible frog *Atelopus zeteki*. *G3: Genes, Genomes, Genetics*, 4(7), 1275-1289.
- Voyles, J., Woodhams, D. C., Saenz, V., Byrne, A. Q., Perez, R., Rios-Sotelo, G., ... & Reinert, L. (2018). Shifts in disease dynamics in a tropical amphibian assemblage are not due to pathogen attenuation. *Science*, 359(6383), 1517-1519.
- Weldon, C., Channing, A., Misinzo, G., & Cunningham, A. A. (2019). Disease driven extinction in the wild of the Kihansi spray toad (*Nectophrynoides asperginis*). *bioRxiv*, 677971.

Reviewers' Comments:

Reviewer #1:

Remarks to the Author:

Greener et al. explore the effects and interactions of low and virulent isolates of *Batrachochytrium dendrobatidis* (Bd) and *B. salamandrivorans* (Bsal) on amphibians. I appreciate the additional experiments that they ran to address concerns that I had with sample sizes. I also appreciate that they dropped most of the presentation of load data from the ms because of the low sample sizes. I think that there is an enormous amount of content in this manuscript that is important to science. For that reason, I encourage publishing it.

While I encourage publishing it, I am torn. For me there is still too much in this ms, making it really hard to digest everything and leaving much of the results not discussed within the context of the broader literature. There just isn't room given the word restrictions and the number of studies they packed into this ms. There is so much in this paper that most of the results are excluded from the abstract, and thus I worry that most of the results will be overlooked. I appreciate the amount that they have accomplished, but my preference is for less content and more detail on the methods and more discussion, even for a short-form, Nature journal article. Nevertheless, I understand that this is personal preference.

As for their argument that they cannot increase sample sizes because of vertebrate animal uses issues, it seems disingenuous to me. I have served on vertebrate animal use committees for years and know these policies extremely well. Yes, the authors did detect significant effects on survival in some experiments, but the authors admit that they had insufficient power to even justify running analyses on Bd loads, which have negative binomial error distributions. Thus, one could very easily justify an increase in animal numbers based on the Bd load endpoint. Additionally, with five animals for some experiments, all it would take is one animal dying to eliminate the significant differences. Again, this could very easily be used to justify additional vertebrate animals. We want to protect against false positives, and additional animals can provide this assurance, especially given the potential costs of implementing a management campaign based on a potential false positive result. So, I found their argument regarding their sample sizes and vertebrate animal restrictions to be far from compelling. So, in many cases, I still find the results based on the low samples sizes to be bit tenuous and preliminary given that a single animal could alter some conclusions (just one experiment now). Nevertheless, the sample sizes are transparent and readers can evaluate how much stock they put in each finding. Thus, I still find considerable value in all the studies that they present.

There are some issues that the authors should address before this is published.

L 48: I think you mean here "not affect by declines". These areas are affected by disease if BdGPL is present and causing symptoms.

L 110: "reflects the European field situation" is vague. What is the European field situation and how do the results reflect this situation?

L 111: I stand by my original assertion that it is ridiculous to claim that the midwife toad is a sentinel of Bd. First off, thank you for providing your definition of sentinel, which is "an indicator of the presence of disease". This only further supports my original assertion. Table S2 provides clear evidence that in several years, certain locations have midwife toads and no detectable Bd. This clearly demonstrates that midwife toads are not a sentinel of the presence of Bd. Moreover, as you so clearly show in this manuscript, the presence of a Bd isolate does not equate to the presence of disease. So, even if midwife toads did reliably indicate the presence of Bd, they might not indicate the presence of disease (an important component of your definition). Thus, your data do not support the conclusions you are drawing in this section of the manuscript for these two reasons.

L 215: This is vague. What would a negative effect have looked like in your field study? I think it was

descriptive and thus couldn't detect a negative effect to begin with.

L 227: Again, vague. What is the focal species? It would not cause declines of alpine newts.

L 247-255: There is no discussion of the results in the context of other studies on amphibian co-infection or acquired resistance, which is disappointing. The author claim in their response to the reviews that they included references to theory/studies on co-infections, but I cannot find any explicit reference to this in the ms. So, it is hard to understand whether this work is consistent or inconsistent with other published work and what elements are novel versus reinforce patterns in the literature. I understand that they simply lack the space, but this is a choice made by the authors because of how much they chose to include in this ms.

L 353-376: Low sample sizes here still. All it takes is one individual to change the significance of the results. Needs to be viewed with some caution.

L 403-433: Thank you for running this more rigorous follow-up experiment.

Reviewer #2:

Remarks to the Author:

Thank you for the opportunity to review this second submission by Greener et al..

As in my initial review, I suggest that the findings of this study are intriguing for several reasons and worthy of consideration for publication. To reiterate my earlier comments, I believe the investigators have disentangled several characteristic features of the molecular and cellular pathophysiology of chytridiomycosis in a way that is novel and highly effective. While previous studies have used similar experimental designs in a piecemeal fashion, this study combines multiple innovative approaches in an effective way and has applied them to multiple potentially vulnerable host species. Because many aspects of this study are impressive in their integration, rigor, and scope, I expect that they will be of interest to a broad readership.

I appreciate that the authors carefully considered my concerns and they have largely addressed the focal issues. In particular, I appreciate that they have done the following:

- Adjusted their argument concerning the question of "exotic" vs "endemic Bd isolates, as well as the terms isolates, strains, and lineages.
- Provided information on the cryopreservation and lab maintenance of all isolates
- Omitted the description of lower virulence isolates providing a "barrier" (but please see comments below)
- Used more cautious language for articulating the importance of their findings for conservation in the discussion.

My only remaining concerns may be addressed with relatively simple modifications:

1) The investigators suggest that infection with a low-virulence isolate provides "protection" against colonization with a more virulent isolate. Their support for this suggestion comes from the sequential exposure of toads to the low pathogenicity isolates and subsequently to the high pathogenicity isolate as well as a similarly designed experiment with salamanders. However, these concerns still remain: The semantics/word choice suggests that pre-exposure provide resistance against subsequent infection (i.e., pathogen colonization). However, from their results, they infer that the toads did become infected with JEL423 (Fig. 3A, right hand column) unless they are supposing that the load results are from the initial exposure (which might occur in a superinfection scenario). If we assume that the load results are from exposure to JEL423, then the toads can clearly become infected with higher pathogenic isolate (JEL423) and at comparable levels of infection intensity. As such, the "protection" could possibly be against development of lethal disease, but pre-exposure does not

protect against infection per se.

This clarification is subtle but important and should be addressed here:

Line 33: "... protects against infection with highly virulent BdGPL...."

Line 60: "...quantified the protective capacity...."

Lines 228-229: "Their protective effect is associated..."

I also still think that the wording is a bit strong given the small sample sizes and mixed results. Given these concerns, I suggest using more careful wording in the Abstract (and explaining the interpretations in the Discussion).

For example:

"Pre-exposure to some of these low virulent isolates MAY confer protection against DISEASE DEVELOPMENT FOLLOWING SUBSEQUENT EXPOSURE TO A highly virulent BdGPL in midwife toads (*Alytes obstetricans*)...."

2) I still think it would be worthwhile to acknowledge the many spatio-temporal factors that may be at play in this system and contributing to lower pathogen virulence in this geographic area (e.g., time since pathogen emergence, local and regional ecological factors, community composition, etc.). Stressing this point would help to avoid the inference that the manifestation of disease is solely due to the inherent pathogenicity of the BdGPL in wild amphibian populations in Europe.

Minor comments:

Given the modifications, I am wondering if the title needs adjustment? It is unclear to me how infection with isolates that are lower in pathogenicity would constitute a "scorched earth" but perhaps I am not understanding what the authors are trying to convey.

The terms "endemic" and "exotic" are still found in the following locations:

Line 174

Figure 3

Supplementary Table 5, Table 9

Line 44: "global" is implied with the term "panzootic"

Line 55: I initially struggled with this sentence. I think the authors mean that they predicted that the Bd in amphibian communities was less pathogenic relative to other Bd isolates? I also suggest breaking this sentence into two: "We explore the extent to which LESS PATHOGENIC isolates MAY confer protection...."

Lines 233-234 Add that this could also set up a superinfection?

Lines 261-265: Awkward sentence

Lines 305: Add parentheses around "2018"

Lines 314: By writing "BdBE1-10", It was not clear how many isolates this is. Perhaps list all the isolates here.

Line 329: Add the year of publication.

Lines 335: between  among

Lines 369 and 428: Ad lib  Ad libitum

We thank the reviewers for their comments. In this rebuttal letter we addressed all issues raised.

Reviewer #1 (Remarks to the Author):

*Greener et al. explore the effects and interactions of low and virulent isolates of *Batrachochytrium dendrobatidis* (Bd) and *B. salamandrivorans* (Bsal) on amphibians. I appreciate the additional experiments that they ran to address concerns that I had with sample sizes. I also appreciate that they dropped most of the presentation of load data from the ms because of the low sample sizes. I think that there is an enormous amount of content in this manuscript that is important to science. For that reason, I encourage publishing it.*

While I encourage publishing it, I am torn. For me there is still too much in this ms, making it really hard to digest everything and leaving much of the results not discussed within the context of the broader literature. There just isn't room given the word restrictions and the number of studies they packed into this ms. There is so much in this paper that most of the results are excluded from the abstract, and thus I worry that most of the results will be overlooked. I appreciate the amount that they have accomplished, but my preference is for less content and more detail on the methods and more discussion, even for a short-form, Nature journal article. Nevertheless, I understand that this is personal preference.

As for their argument that they cannot increase sample sizes because of vertebrate animal uses issues, it seems disingenuous to me. I have served on vertebrate animal use committees for years and know these policies extremely well. Yes, the authors did detect significant effects on survival in some experiments, but the authors admit that they had insufficient power to even justify running analyses on Bd loads, which have negative binomial error distributions. Thus, one could very easily justify an increase in animal numbers based on the Bd load endpoint. Additionally, with five animals for some experiments, all it would take is one animal dying to eliminate the significant differences. Again, this could very easily be used to justify additional vertebrate animals. We want to protect against false positives, and additional animals can provide this assurance, especially given the potential costs of implementing a management campaign based on a potential false positive result.

So, I found their argument regarding their sample sizes and vertebrate animal restrictions to be far from compelling. So, in many cases, I still find the results based on the low samples sizes to be bit tenuous and preliminary given that a single animal could alter some conclusions (just one experiment now). Nevertheless, the sample sizes are transparent and readers can evaluate how much stock they put in each finding. Thus, I still find considerable value in all the studies that they present.

There are some issues that the authors should address before this is published.

L 48: I think you mean here "not affect by declines". These areas are affected by disease if BdGPL is present and causing symptoms.

disease is replaced by declines

L 110: "reflects the European field situation" is vague. What is the European field situation and how do the results reflect this situation?

*to clarify we added '(endemic and widespread presence of *Bd*, with focal outbreaks)' to the sentence*

L 111: I stand by my original assertion that it is ridiculous to claim that the midwife toad is a sentinel of Bd. First off, thank you for providing your definition of sentinel, which is "an indicator of the presence of disease". This only further supports my original assertion. Table S2 provides clear evidence that in several years, certain locations have midwife toads and no detectable Bd. This clearly demonstrates that midwife toads are not a sentinel of the presence of Bd. Moreover, as you

so clearly show in this manuscript, the presence of a *Bd* isolate does not equate to the presence of disease. So, even if midwife toads did reliably indicate the presence of *Bd*, they might not indicate the presence of disease (an important component of your definition). Thus, your data do not support the conclusions you are drawing in this section of the manuscript for these two reasons.

We think we now understand the concern of this reviewer: we did not mean to propose midwife toads to be suitable sentinels for the occurrence of *Bd*. What we mean is that midwife toads are a sentinel species for virulent *Bd* infections (not only our results but all available literature data corroborate this statement), what we associate with the occurrence of highly virulent *Bd* GPL isolates. We clarified this as follows:

We replaced “(indicator of the presence of disease)” by (canary in the coal mine for virulent *Bd* infections in Europe)

L 215: This is vague. What would a negative effect have looked like in your field study? I think it was descriptive and thus couldn't detect a negative effect to begin with.

The following conclusions are sound based on 4 years of follow up in midwife toads and a seasonal follow up in Alpine newts:

Midwife toad populations (which are all remnant, small populations (added to materials and methods)) showed no signs of population crashes in the presence of *Bd*; *Bd* was shown to co-exist with midwife toad populations for at least four years.

Infected Alpine newt populations were indistinguishable from uninfected populations with regard to newt abundance and body condition during the breeding season.

We do agree with this reviewer that the statement may need some specification and changed: “Our field surveys did not uncover any negative impact of *Bd* endemism in wild populations of midwife toads and alpine newts.” to: “Midwife toads persisted in small and isolated populations during at least four years in the presence of *Bd* and *Bd* infection could not be linked to decreased body condition or host abundance in alpine newts.”

L 227: Again, vague. What is the focal species? It would not cause declines of alpine newts.

We replaced “focal” by “susceptible”

L 247-255: There is no discussion of the results in the context of other studies on amphibian co-infection or acquired resistance, which is disappointing. The author claim in their response to the reviews that they included references to theory/studies on co-infections, but I cannot find any explicit reference to this in the ms. So, it is hard to understand whether this work is consistent or inconsistent with other published work and what elements are novel versus reinforce patterns in the literature. I understand that they simply lack the space, but this is a choice made by the authors because of how much they chose to include in this ms.

We have indeed been restrained by word and reference limitations. The only relevant study we are aware of, is the study of Longo et al. (2018). We sacrificed Lötters et al., 2018, to include Longo et al. instead. Please note that this study has a very different setup and premise (simultaneous co-infections), which very much limits relevance for the current epidemiological situation. We added the sentence:

In amphibian communities, simultaneous introduction of both pathogens may be subject to different disease dynamics. Simultaneous co-infections with *Bd* and *Bsal* in American newts (*Notophthalmus viridescens*) resulted in disease exacerbation⁴³.

L 353-376: Low sample sizes here still. All it takes is one individual to change the significance of the results. Needs to be viewed with some caution.

We would like to refer to our first revision. We fundamentally disagree with this reviewer. We are equally involved in several ethical committees on animal experimentation in Europe and we can assure that using a minimum of animals, yet reaching meaningful results, has become insurmountable. We feel we did not oversell the results (we will not conclude that there is no impact of *Bd* on these other species at population level based on an experiment with 5 (actually 10) animals per species, which would indeed not be justified).

L 403-433: Thank you for running this more rigorous follow-up experiment.

Reviewer #2 (Remarks to the Author):

Thank you for the opportunity to review this second submission by Greener et al..

As in my initial review, I suggest that the findings of this study are intriguing for several reasons and worthy of consideration for publication. To reiterate my earlier comments, I believe the investigators have disentangled several characteristic features of the molecular and cellular pathophysiology of chytridiomycosis in a way that is novel and highly effective. While previous studies have used similar experimental designs in a piecemeal fashion, this study combines multiple innovative approaches in an effective way and has applied them to multiple potentially vulnerable host species. Because many aspects of this study are impressive in their integration, rigor, and scope, I expect that they will be of interest to a broad readership.

I appreciate that the authors carefully considered my concerns and they have largely addressed the focal issues. In particular, I appreciate that they have done the following:

- Adjusted their argument concerning the question of “exotic” vs “endemic *Bd* isolates, as well as the terms isolates, strains, and lineages.
- Provided information on the cryopreservation and lab maintenance of all isolates
- Omitted the description of lower virulence isolates providing a “barrier” (but please see comments below)
- Used more cautious language for articulating the importance of their findings for conservation in the discussion.

My only remaining concerns may be addressed with relatively simple modifications:

1) The investigators suggest that infection with a low-virulence isolate provides “protection” against colonization with a more virulent isolate. Their support for this suggestion comes from the sequential exposure of toads to the low pathogenicity isolates and subsequently to the high pathogenicity isolate as well as a similarly designed experiment with salamanders. However, these concerns still remain:

The semantics/word choice suggests that pre-exposure provide resistance against subsequent infection (i.e., pathogen colonization). However, from their results, they infer that the toads did become infected with JEL423 (Fig. 3A, right hand column) unless they are supposing that the load results are from the initial exposure (which might occur in a superinfection scenario). If we assume that the load results are from exposure to JEL423, then the toads can clearly become infected with higher pathogenic isolate (JEL423) and at comparable levels of infection intensity. As such, the “protection” could possibly be against development of lethal disease, but pre-exposure does not protect against infection per se.

This clarification is subtle but important and should be addressed here:

Line 33: “... protects against infection with highly virulent *Bd*GPL...”

Line 60: “...quantified the protective capacity....”

Lines 228-229: “Their protective effect is associated...”

I also still think that the wording is a bit strong given the small sample sizes and mixed results. Given

these concerns, I suggest using more careful wording in the Abstract (and explaining the interpretations in the Discussion).

For example:

“Pre-exposure to some of these low virulent isolates MAY confer protection against DISEASE DEVELOPMENT FOLLOWING SUBSEQUENT EXPOSURE TO A highly virulent BdGPL in midwife toads (*Alytes obstetricans*)....”

We agree with the argument that protection should be specified. Indeed, this is protection against clinical disease, not infection. We changed the wording accordingly:

Line 33: “... protects against infection with highly virulent *BdGPL*....” changed to: “Pre-exposure to some of these low virulent isolates protects against disease following subsequent exposure to highly virulent *BdGPL*” We would prefer not to use “may” here, since the additional experiment independently confirms previous results in a larger number of animals.

Line 60: “...quantified the protective capacity....” changed to: “Finally, we estimated the impact of pre-existing infections with less virulent *BdGPL* isolates on virulent *BdGPL* and *Bsal* infections.”

Lines 228-229: “Their protective effect is associated...”, we added “against disease”

2) I still think it would be worthwhile to acknowledge the many spatio-temporal factors that may be at play in this system and contributing to lower pathogen virulence in this geographic area (e.g., time since pathogen emergence, local and regional ecological factors, community composition, etc.). Stressing this point would help to avoid the inference that the manifestation of disease is solely due to the inherent pathogenicity of the *BdGPL* in wild amphibian populations in Europe.

We agree with the author here. It is not our intention to ascribe disease dynamics to the presence of highly virulent isolates only and tried to make that clear throughout the manuscript (eg in the abstract: “The key role of pathogen virulence in the complex host-pathogen-environment interaction that drives infection and disease dynamics...” or further in the discussion: “In a suitable environment, these disease dynamics might result in mass die-offs, population declines and even extirpation of the susceptible species³⁸.”

We suggest to make this a bit more explicit by altering that latter sentence into: “If spatio-temporal factors shape a conducive environment, these disease dynamics might...”

Minor comments:

Given the modifications, I am wondering if the title needs adjustment? It is unclear to me how infection with isolates that are lower in pathogenicity would constitute a “scorched earth” but perhaps I am not understanding what the authors are trying to convey.

We agree with this reviewer and with the editor. We slightly changed the suggestion of the editor and propose the title:

Presence of low virulence chytrid fungi could protect European amphibians from more deadly strains

The terms “endemic” and “exotic” are still found in the following locations:

Line 174

Figure 3

Supplementary Table 5, Table 9

The terms endemic and exotic were changed.

Line 44: “global” is implied with the term “panzootic”

global is deleted

Line 55: I initially struggled with this sentence. I think the authors mean that they predicted that the

Bd in amphibian communities was less pathogenic relative to other Bd isolates? I also suggest breaking this sentence into two: “We explore the extent to which LESS PATHOGENIC isolates MAY confer protection...”

Please see comment above. We changed this sentence to: “Finally, we estimated the impact of pre-existing infections with less virulent *Bd*GPL isolates on virulent *Bd*GPL and *Bsal* infections.”

Lines 233-234 Add that this could also set up a superinfection?

Possibility of superinfection is added

Lines 261-265: Awkward sentence

We changed this sentence to: “In a first study, *Bd* prevalence was determined across our study area (Flanders, Belgium). We sampled 1,483 amphibians belonging to 62 populations in 2015-2016 (Supplementary Figure 1).”

Lines 305: Add parentheses around “2018”

Parentheses are added

Lines 314: By writing “BdBE1-10”, It was not clear how many isolates this is. Perhaps list all the isolates here.

All the isolates are listed

Line 329: Add the year of publication.

2009 is added

Lines 335: between  among

Between is changed into among

Lines 369 and 428: Ad lib  Ad libitum

ad lib is changed into ad libitum in the text